# Emergence of corpse cremation during the Pre-Pottery Neolithic of the Southern Levant: A multidisciplinary study of a pyre-pit burial

Fanny Bocquentin[1]*, Marie Anton[2,3], Francesco Berna[4], Arlene Rosen[5], Hamoudi Khalaily[6], Harris Greenberg[7], Thomas C. Hart[8], Omri Lernau[9], Liora Kolska Horwitz[10]

1 Cogitamus Laboratory and CNRS, UMR 7041, ArScAn, Equipe Ethnologie Préhistorique, MSH Mondes, Nanterre, France, 2 Université Paris 1, Panthéon-Sorbonne, Paris, France, 3 CNRS, UMR 7206, Musée de l'Homme, Éco-Anthropologie et Ethnologie, Paris, France, 4 Department of Archaeology, Simon Fraser University, Burnaby, Canada, 5 Department of Anthropology, University of Texas at Austin, Austin, TX, United States of America, 6 Israel Antiquities Authority, Jerusalem, Israel, 7 Department of Archaeology, Boston University, Boston, MA, United States of America, 8 Department of Anthropology, Franklin and Marshall College, Lancaster, PA, United States of America, 9 Zinman Institute of Archaeology, University of Haifa, Haifa, Israel, 10 National Natural History Collections, The Hebrew University, Jerusalem, Israel

* fanny.bocquentin@cnrs.fr

**Data Availability Statement:** All supplementary files are available from the Nakala database. S1:

## Abstract

Renewed excavations at the Neolithic site of Beisamoun (Upper Jordan Valley, Israel) has resulted in the discovery of the earliest occurrence of an intentional cremation in the Near East directly dated to 7031–6700 cal BC (Pre-Pottery Neolithic C, also known as Final PPNB, which spans ca. 7100–6400 cal BC). The funerary treatment involved *in situ* cremation within a pyre-pit of a young adult individual who previously survived from a flint projectile injury. In this study we have used a multidisciplinary approach that integrates archaeothanatology, spatial analysis, bioanthropology, zooarchaeology, soil micromorphological analysis, and phytolith identification in order to reconstruct the different stages and techniques involved in this ritual: cremation pit construction, selection of fuel, possible initial position of the corpse, potential associated items and funerary containers, fire management, post-cremation gesture and structure abandonment. The origins and development of cremation practices in the region are explored as well as their significance in terms of Northern-Southern Levantine connections during the transition between the 8th and 7th millennia BC.

## Introduction

The treatment of the dead during the Neolithization process in the Near East was a complex process, embedded in a cognitive and symbolic world that underpinned the economic and dietary shift from hunting and gathering to agro-pastoralism [e.g. articles in 1]. Burial location and funerary gestures varied from one community to another as well as between one individual and another within the same site. Thus, several operational sequences in burial practice co-existed resulting in primary, secondary, plural, single, staging, manipulations and/or skeletal

https://doi.org/10.7794/p6w6-f483 S2: https://doi.org/10.7794/13jm-9k62.

**Funding:** The Irene Levi Sala Care Archaeological Foundation supported financially this research project untitled Domesticating decay during the Pre-Pottery Neolithic: emergence of corpse cremation in Southern Levant (F. Bocquentin and A. Rosen Dir.). The soil micromorphology analyses were partially funded by the Canada Social Sciences and Humanities Research Council (Grant No. 435-2016-1123). We are also indebted to our host institutes and financial supporters which make possible the renewed excavation program at Beisamoun: French National Center of Scientific Research (CNRS, UMR7041, Ethnologie Préhistorique), Israel Antiquities Authority (IAA), French center of Research at Jerusalem (CRFJ), French Ministry of Foreign Affairs (MEAE).

**Competing interests:** The authors have declared that no competing interests exist.

element removal that were carried out, sometimes side by side. Occasional defleshing, and/or dismemberment, and temporary mummification are also suspected to have been practiced. The variety of treatments has its roots in the preceding Natufian culture (ca. 13000–9600 cal BC), and possibly even before, but becomes more complex and diversified throughout the Pre-Pottery Neolithic period (ca. 9600–6400 cal BC) [for some recent examples: 2–12].

The practice of cremation has also been mentioned several times for pre-ceramic periods in the Near East but often questioned, since accidental fire exposure cannot, in most instances, be ruled out. The presence of burnt isolated human bones is not exceptional but remains poorly understood due to the lack of accurate contextual and taphonomic studies [see synthesis in: 13]. It seems that fire was an integral part of funerary ceremonies as hearths were regularly situated close to, or were in association with graves at least from the beginning of sedentarization [e.g. 8, 14–18]. However, intentional exposure of the corpse to fire is a step that these pre-ceramic Near Eastern communities do not seem to have taken, despite a unique instance from the Early Natufian occupation at Kebara Cave [19]. Here the bioanthropological analysis demonstrated secondary deposition of burnt bones belonging to a minimum of 31 individuals. Of these, only one may represent a primary cremation, the others, probably having been burnt after the decay process (colouration of the bones favours burning at a low temperature, in a fire that was not tended [16: 151–160]). This isolated Epipaleolithic case remains equivocal and difficult to interpret.

Currently, the earliest well-established cases of intentional fire-induced modification from the region date to the Neolithic and involved dry bones and not fresh corpses [8, 20–23]. Although, within bioanthropological communities "cremation" usually refers to corpse-burning, in the Near Eastern context, intentional burning of dry bones may also qualifies as "cremation practices" [3, 7, 20–21]. Given their distinct characters, we should probably differentiate between the two. We refer to them here using the terms "primary cremation" for corpse burning and "secondary cremation" for burning of dry remains.

In both cases, the process of intentional burning involves a complex technical protocol (requiring fuel, oxygen circulation and high temperature), which is partly recognizable in archaeological contexts [e. g. 24–27]. The incineration of a corpse requires combustion at a high temperature (>600˚), for several hours [e.g. 25, 28, 29]. The more the combustion is tended (constant heat, segments of corpse recentered in the furnace) the faster and more complete the process [e.g. 24, 28–31]. Thus, the intention can be determined by recognition of features such as pyre structures, intensity and extent of burning to the bones, their positioning and context as well as the repetition of such cases.

In this paper we present the earliest occurrence of an intentional primary cremation from the Near East known to date, from the site of Beisamoun in Northern Israel. It places the emergence of this practice to a critical transition period, between the Late Pre-Pottery Neolithic B (PPNB) and the Pre-Pottery Neolithic C (PPNC also known as the Final PPNB), on the eve of the 7th millennium BC.

# 1. Beisamoun: Context and field work

## 1.1 The site and its chrono-cultural context

Beisamoun is a Pre-Pottery Neolithic settlement located on the shores of the ancient Hula Lake (Upper Jordan Valley, Israel) (Fig 1). It has been shown to be a key site for understanding the PPNB and PPNC, the terminal phase of the Pre-Pottery Neolithic in the Southern Levant, since it was occupied during the 8th and 7th millennia BC. We know from previous research and from deep trenches dug by us, that the site was occupied from at least the Middle PPNB [32–34]. Renewed excavations from 2007 to 2016, under the direction of F. Bocquentin and H.

Khalaily, revealed the very end of the occupation in the PPNC before site abandonment [33–37]. Thus, Beisamoun provides an uninterrupted occupational sequence covering the poorly known transition between the Late PPNB and the Early Pottery Neolithic in the Southern Levant.

In contrast to the Northern Levant, pottery is not attested in the Southern Levant before 6400–6200 cal BC and there is a clear break in the interaction sphere that linked the Northern and Southern Levant throughout the PPNB, via cultural exchange and possible population movements [e.g. 38, 39]. At the end of the Late PPNB, many of the large villages (megasites) were abandoned [40]. This so-called "Palestinian hiatus", long considered a gap, is now better known thanks to recent excavations at key sites, such as 'Ain Ghazal, Wadi Shueib, Beisamoun and most recently Motza [41], whose occupation continues throughout the PPNC.

At Beisamoun, a continuous occupation, at least from 7200 Cal BC up to 6400/6200 Cal BC, is attested by radiocarbon dates in the sector recently excavated [37]. An intensive period of building and rebuilding characterized the onset of the 7th millennium. Stratigraphic, architectural, material production and symbolic continuity with the Late PPNB is observed, at least until ca. 7500 Cal BC. After this, the site shows major changes with lighter construction, and the disappearance of intramural burials. This constitutes a major evolutionary shift before the end of the PPNC, following which the site was abandoned [37].

In the LPPNB/PPNC layers at the site, a minimum of 33 individuals were identified, including 18 adults and 15 immature individuals (12 are infants). There is a great diversity of funerary practices at this time, with single or double graves, including 19 primary burials, 11

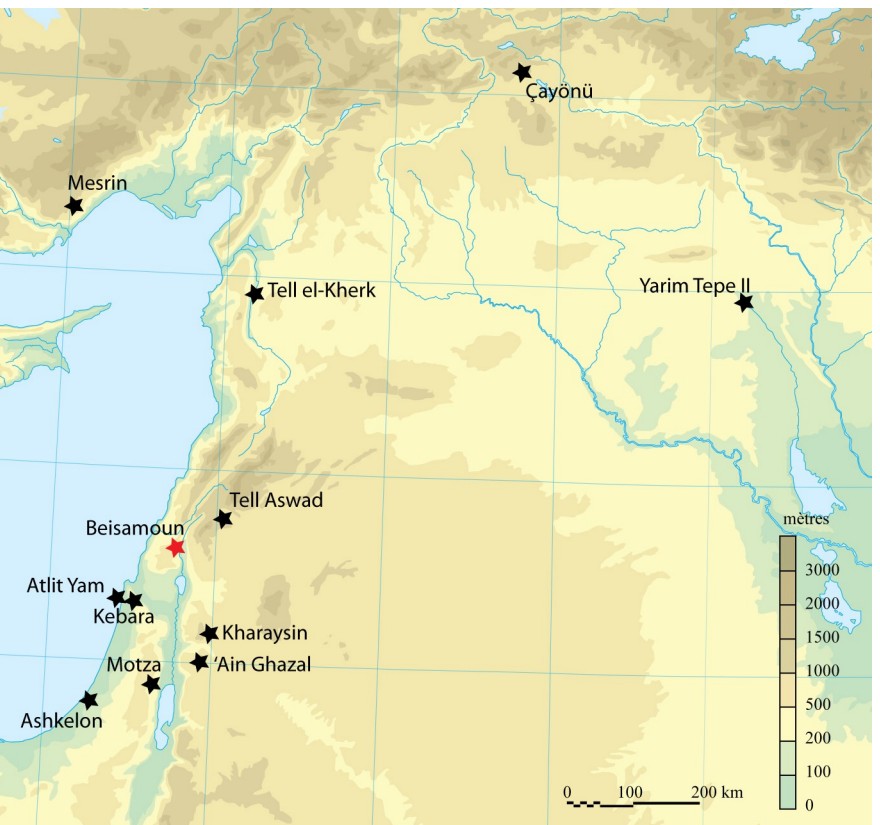

**Fig 1. Map of sites mentioned in the text (ground map: M. Sauvage, copyright CNRS).**

secondary burials (five are secondary deposits of cremated remains), two undefined deposits and a primary cremation in a pyre-pit (Locus 338), which is the subject of this article. Practices of cranium removal and plastering are attested at Beisamoun in the transition period between Late PPNB and PPNC ($^{14}$C dated here ca 7300/7200-7100/7000 Cal BC)[34, 37], and seem to be the latest attestation of these iconic practices inherited from earlier Pre-Pottery Neolithic periods [32]. The appearance of cremation at approximately the same period constitutes a major shift in burial practice and signifies a clear break with the preceding Late PPNB.

## 1.2 Excavation process of Locus 338

The cremation in the pyre-pit — Locus 338, as all other features excavated at Beisamoun [see 34], were dug in successive spits using the *décapages* method, which consists of exposing and mapping finds as they appear, level by level, using a unique bank of catalogue numbers. For the pyre-pit, each piece of bone received a specific number and was mapped using three co-ordinates (x-y-z) and drawn to scale *in situ*. We first exposed the top circumference of the pit entirely. As the pit was partially eroded to the north, we excavated extremely carefully in this area before reaching the preserved pit top. Once its perimeter was entirely defined, we divided the pit into four sectors in line with the overall excavation grid. In order to understand the pit fill, the south-east quarter which was preserved to the greatest height, was first explored to a depth of ca. 25 cm in four successive spits (catalogue #'s 2791, 2804, 2933, 2969). Spit # 2804 contained the greatest abundance of calcined bones within the pit fill (photographed *in situ* in Fig 2B). We then explored the fill of the north-east part (catalogue # 2823), and of the north-west part (catalogue # 2879). Having reached a small bench that occupies a large part of the north-west quarter (i.e. a part of the pit wall that juts out and is ca. 20 cm wide), we continued to explore the northern half of the pit (catalogue #s 2904, 2919). Finally, we explored the south-western quarter of the pit in four successive spits (#s 2975, 2995, 3007, 3015). At this stage, having reached the same spit height throughout the structure, we continued to expose and dismantle the bones in the deepest part, through 11 successive spits (catalogue #s 3051, 3102, 3114, 3142, 3165, 3176, 3190, 3197, 3205, 3209, 3210) (S1 Fig). All the necessary permits for this study have been obtained according to the regulations. The human remains involved (Locus 338) were unearthed under the field permit number: G39-2012; G41-2013 issued by the Israeli Antiquities Authority (IAA). They are currently kept at the French National Research Center in Jerusalem (CRFJ) and they will be handed over to the IAA after publication. In the meantime, they are available for review upon request to the CRFJ and the corresponding author of the current article.

## 1.3 Archaeothanatology

We have applied here an archaeothanatological approach (i.e. taphonomic analysis of burials) to the study of the human remains in Locus 338. This field of research lies at the intersection between bioanthropology and funerary archaeology and is today a well-developed area of research worldwide [e.g. 42–48]. It deals with the operational chain of corpse and funeral processing [e.g. Ibid and 49, 50] and highlights the relation between the dead and living through fine stratigraphic analysis [e.g. 51].

Archaeothanatology examines the final spatial distribution of human remains as seen during excavation in relation to the initial setting of a burial. Indeed, the preservation, or to the contrary, the movement of skeletal elements and joints, permits the reconstitution of the initial placement of the corpse and which elements were present in the grave at the time of burial. Despite the fact that the timing of putrefaction is both context-specific and highly variable, joints do not decay haphazardly and the relative sequence of their dislocation is known [e.g.

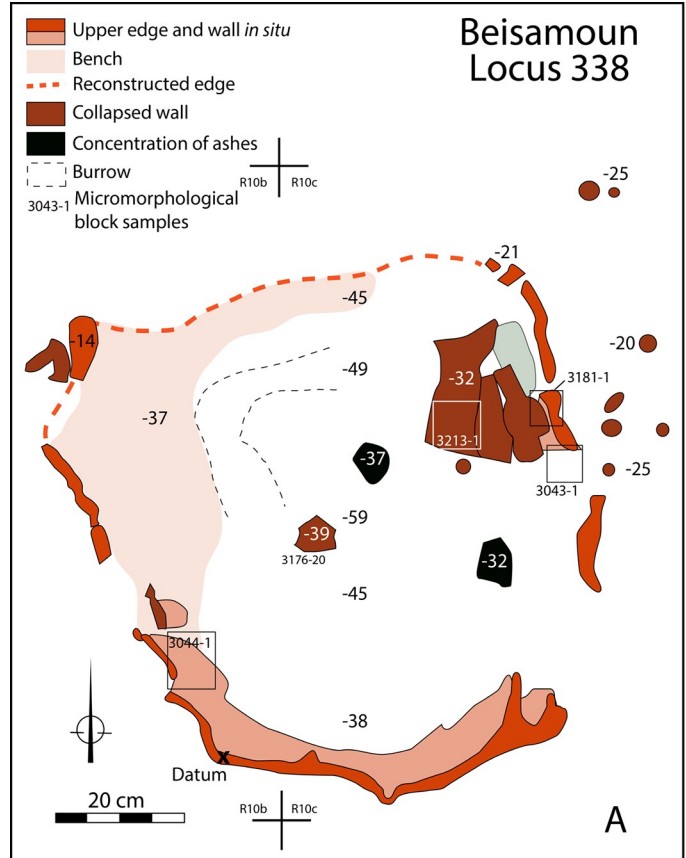

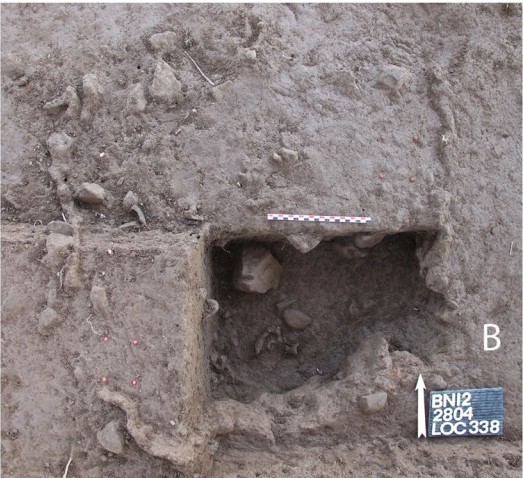

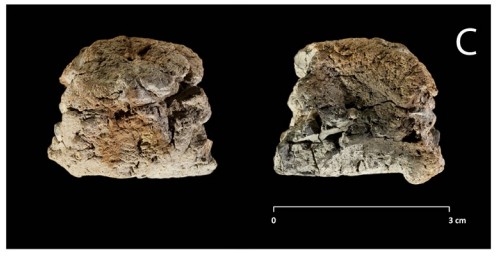

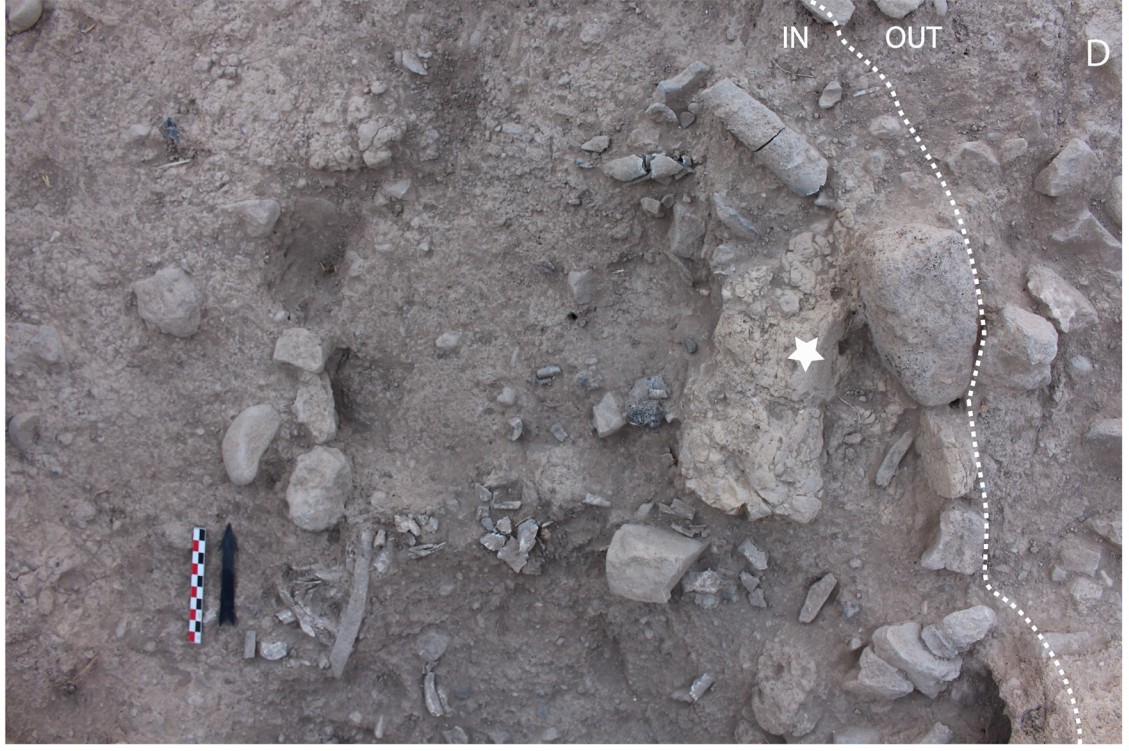

**Fig 2. Structure 338.** A: Map of the cremation pyre-pit; B: Identification of the upper edges of the pit and ongoing excavation of its south-east fourth. C: Details of a piece of mud-plastered wall (ref 3142–12; copyright L. Teira). D. Detail of the north-east quarter of the pit during excavation. A fragment of collapsed wall inside the pit is visible (star).

42, 46]. In the context of primary burial, if the connections between the labile joints are preserved (e.g. bones of the hand or feet), it is unlikely that the skeleton was disturbed shortly after death. To the contrary, if articulations that are more enduring are found disarticulated, this may provide evidence for their movement long after death. Through rigorous spatial analysis of the skeletal remains, burial position, evidence of wrappings/containers, the physical relation between the corpse with other funerary features (pillow, bed, faunal offerings etc.) and other objects in the grave as well as post-depositional intervention, can be identified [e.g. *princeps* articles: 52, 53].

In the context of pyres and cremated bones, archaeothanatology appears to be the most effective approach to reconstruct the sequence of events, together with physical alterations evident on the bones (colour, warping, cracks, fragmentation etc.). Spatial organization of cremated bones provides crucial information on pyre management, state of the body when burnt (fresh, mummified, skeletonized, dry disarticulated bones) and possibly more (corpse orientation and position at deposit, funerary *apparatus*) [e.g. 27, 30, 31, 54]. These investigations of context are complemented by data on heat signatures which, by themselves, are often equivocal [e.g. 27, 55, 56].

At Beisamoun, particular attention was paid to the spatial distribution of the bones found in Locus 338 and their anatomical relationship. In the case of joint contiguity, the observations on whether they were articulated or not, was scrupulously observed. In addition, laboratory analysis resulted in a comprehensive inventory including anatomical identification, classification of heat alterations (colour, warping etc.) and weight of each piece of bone found in the pyre (S1 Table).

## 2. Construction of the pyre-pit

Locus 338 is located in Sector E of the site (Squares R10 and R11). It is a U-shaped pit that is 80 cm in its largest diameter and 60 cm deep (Fig 2A and 2B). It was dug down from Layer Ib into Layer Ic. Layer Ib corresponds to a collapse of mud brick (that belonged to the walls of the earlier phase—Layer Ic) and before reconstruction of the next phase (Layer I). At the time of the pyre-pit excavation, the area seems to have been temporarily abandoned and was dedicated to burial and ritual activities as witnessed by the presence of ceremonial arrangements (platforms, ritual animal deposits) and graves dug inside abandoned houses near the partially ruined walls [37].

The walls of the pyre-pit were ca. 2 cm thick, indurated and reddish in colour and so were easily identified during the excavation, but brittle once exposed. The eastern and southern pit edges were wider in extent before sinking vertically. As noted above, the western edge had a small bench. The northern edge of the pit was not well preserved, but the distribution of the bones and their steep dip make it possible to reconstruct the outline of this wall with sufficient certainty (dashed line on drawing Fig 2A). The appearance of the wall is irregular both in shape and thickness. Sediment micromorphological analysis revealed that the wall of the pit was plastered with a 1 to 3 cm-thick carbonatic mud-based material (Fig 2C) resembling the raw material used for some of the mud bricks excavated at the site [34]. Vesicular micro-porosity and iron depletion pedofeatures indicate that this plastering material had been applied wet to the walls of the pit (Fig 3) and eventually fired *in situ*.

## 3. The deceased

### 3.1 Skeletal elements

A total of 355 fragments of identified human bone were unearthed from the cremation pit where they were encased in a fine ashy sediment. All parts of the skeleton are represented although they differ in frequency i.e. teeth, carpals and tarsals are scarce. No duplication of skeletal elements exists and the Minimal Number of Individuals was estimated as one, identified as a young adult (Fig 4). The age is based on the fact that all preserved epiphysis are fused, but vertebrae S2 and S3 are only partially fused (fusion before 30 years old see [57]). Sex could not be estimated with confidence as the bones are fragmented and mostly calcinated. The vertebrae have remodelled joint surfaces compatible with moderate arthritis.

The total weight of this bone assemblage is 1158.7 g which reinforces that only a single individual was present because the individual reference weight averages (for both a female and *a fortiori* for a male) are higher than the weight of this cremation [e.g.: 29, 58]. The removal of a few bones at the end of cremation cannot be excluded, particularly for the least represented skeletal elements (see Fig 4 and S1 Table), but difficulties of identification due to fragmentation may also be a factor [30, 58].

The calcinated right fibula of the cremation was directly dated to 7031–6700 cal BC (Beta-514789; 7950±30 BP; 95,4% of confidence BetaCal3.21) such that it is clearly not intrusive into the site and is contemporaneous with Layer Ib [37].

### 3.2 The embedded projectile point

The young deceased had been the victim of interpersonal violence, but survived. A fragment of a projectile point (of 11.6 mm length preserved) was still embedded in the base of the left scapular spine (preserved piece of bone measure 32.82 mm in length). The perforated bone remodelled around the flint fragment, which had been broken by bending at both ends (Fig 5B–5F). The orientation of the projectile shows that it penetrated through the posterior surface of the bone from the top left towards the *infraspinous fossa* (Fig 5A). It perforated the entire width of the scapular spine. Depending on the position of the victim and the movement of the arm at the time of impact, the shooter was positioned either to the back or on the left side of the deceased. The *infraspinatus* muscle was possibly torn, which most likely caused a large hematoma and severe pain but not necessarily impaired function. The individual survived the injury, based on the completely healed area of the wound, which may take 6 weeks to a few months to heal [59, 60]. The area around the wound does not show specific signs of infection (Fig 5C–5H). The projectile tip remained embedded in the shoulder until the individual's death (a few months or few years after the injury). Because of cremation, the flint exhibits pot lid fractures caused by heating and a change of colour (Fig 5C and 5D). The broken tip (0.5–1.5 mm) is not present in the remodelled bone and must have fall into the pyre but was not identified despite fine sieving.

### 3.3 Spatial analysis of the human remains

The bones are distributed throughout the bottom of the pit, partly superimposed one on the other to a thickness of 40 cm. However, the density of remains was not very marked except at the centre of the pit (Fig 6). If there was an apparent anatomical disorder at first glance, by looking at the details some interesting patterning could be observed. Cranial and mandibular fragments were found only in the southern half of the structure. Next to the south wall on the upper level, we found the base of the skull (mandible reversed and occipital fragments); the rest of the cranial vault and face (frontal, maxillars, parietals and temporals) were found

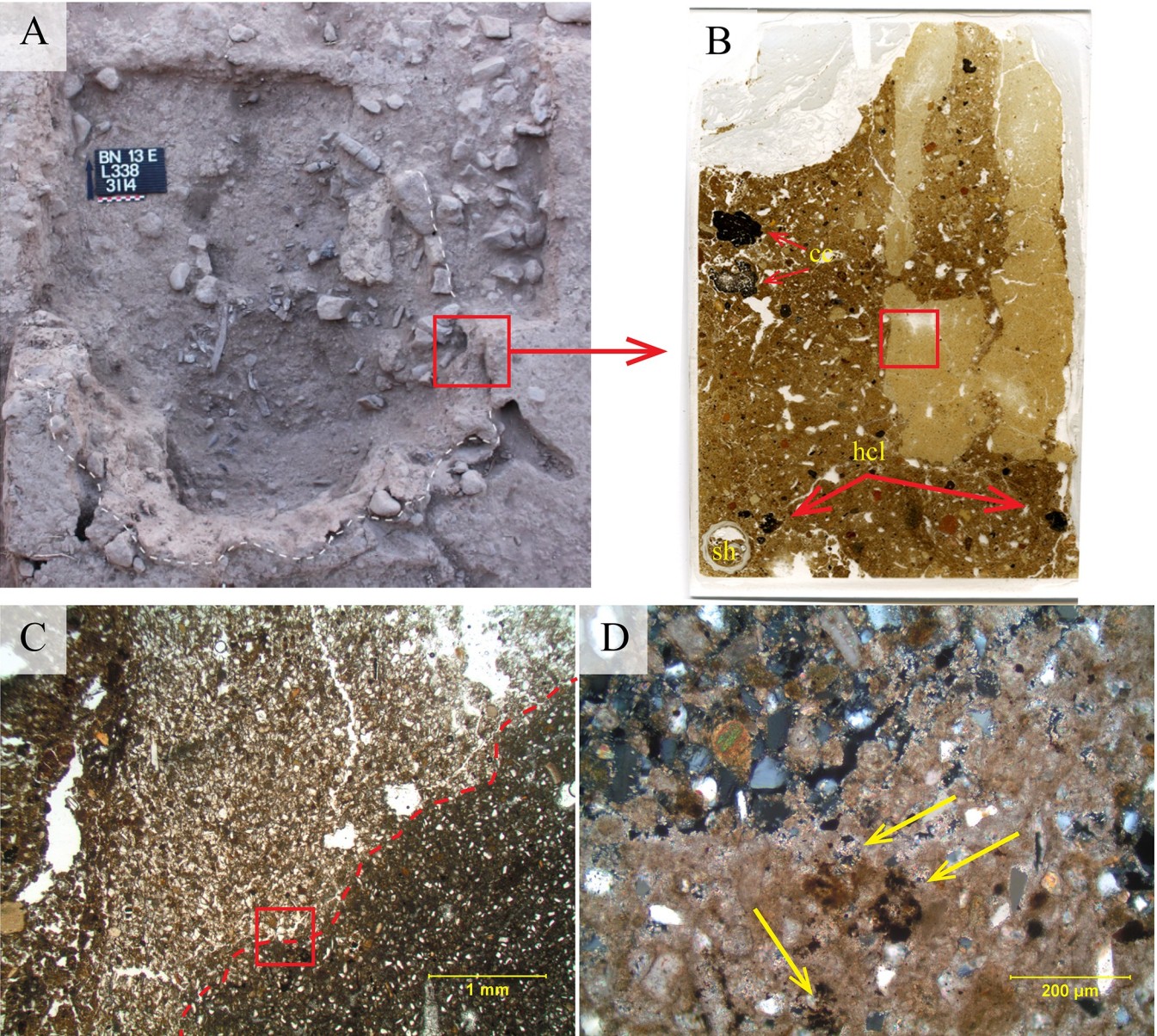

**Fig 3. Micromorphology of the exogenous material used to plaster the cremation pit.** A) Photograph of Locus 338 and the sampling location (red box) of the plastered wall. B) Macroscan of the 50x75mm petrographic thin section processed from the intact block of *in situ* plaster sampled from the area indicated by the red box in Fig 3A. Note the groundmass with spongy peds of reddish-yellow (7.5YR 6/6) to very pale brown (10YR 7/4) clay with common silt to very fine, sand-sized grains of quartz and calcite. cc = heated limestone. Hcl = heated terra rossa soil aggregates. Sh = gastropod shell. C) Microphotograph of the *in situ* contact between the plastered wall surface of Locus 338 and the sediment matrix filling the pit. D) Microphotograph of area (red block in photo C) of well-preserved carbonatic mud plaster from the *in situ* wall of Locus 338. Note the massive light grey (10YR 7/1) matrix of calcium carbonate, with frequent (30–50% of field of view) silt-sized (2–20 μm) grains of quartz and calcite. Mixed into the calcium carbonate matrix are small fragments of weathered bone, shell, phytoliths, basalt fragments, and charcoal. Portions of the structure are slightly decalcified. The exterior edge appears to be physically degraded, so any burnishing or surface decoration is not preserved.

slightly lower down at the centre of the pit. Conversely, the cervical vertebrae were dispersed out from the centre to the northern half of the pit. The thoracic column and some of the ribs were concentrated in the centre, roughly following a west-east direction. The lumbar vertebrae were found in the middle and against the south-western wall of the structure with several vertebral fragments in close proximity to the sacrum, coccyx and the left coxal. The

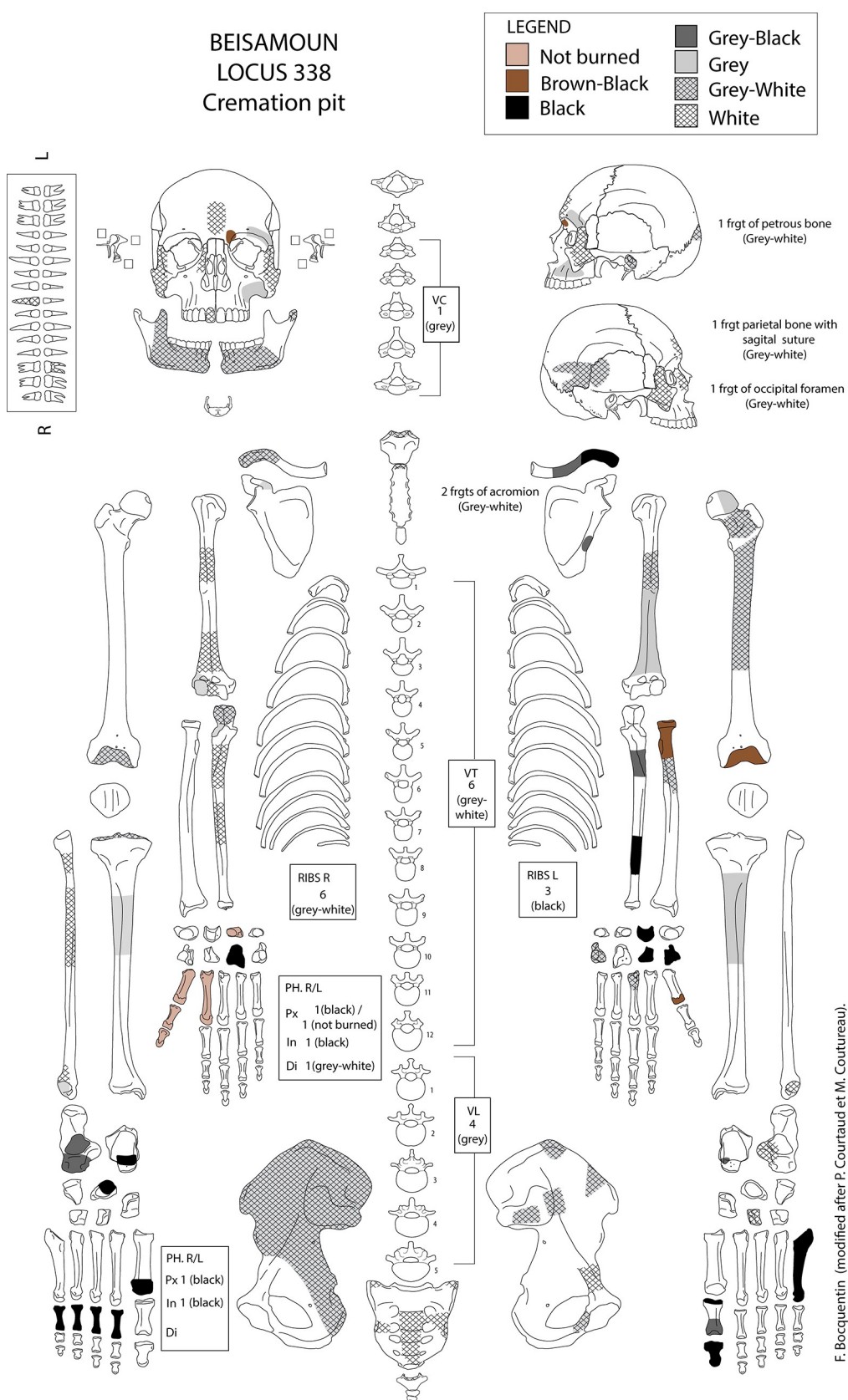

**BEISAMOUN
LOCUS 338
Cremation pit**

LEGEND
Not burned
Brown-Black
Black
Grey-Black
Grey
Grey-White
White

1 frgt of petrous bone
(Grey-white)

1 frgt parietal bone with
sagital suture
(Grey-white)

1 frgt of occipital foramen
(Grey-white)

2 frgts of acromion
(Grey-white)

VC
1
(grey)

VT
6
(grey-
white)

VL
4
(grey)

RIBS R
6
(grey-white)

RIBS L
3
(black)

PH. R/L

Px    1(black) /
        1 (not burned)
In    1 (black)
Di    1(grey-white)

PH. R/L
Px 1 (black)
In 1 (black)
Di

F. Bocquentin  (modified after P. Courtaud et M. Coutureau).

**Fig 4. Inventory sheet of the cremated individual.** Coloration of bones are specified.

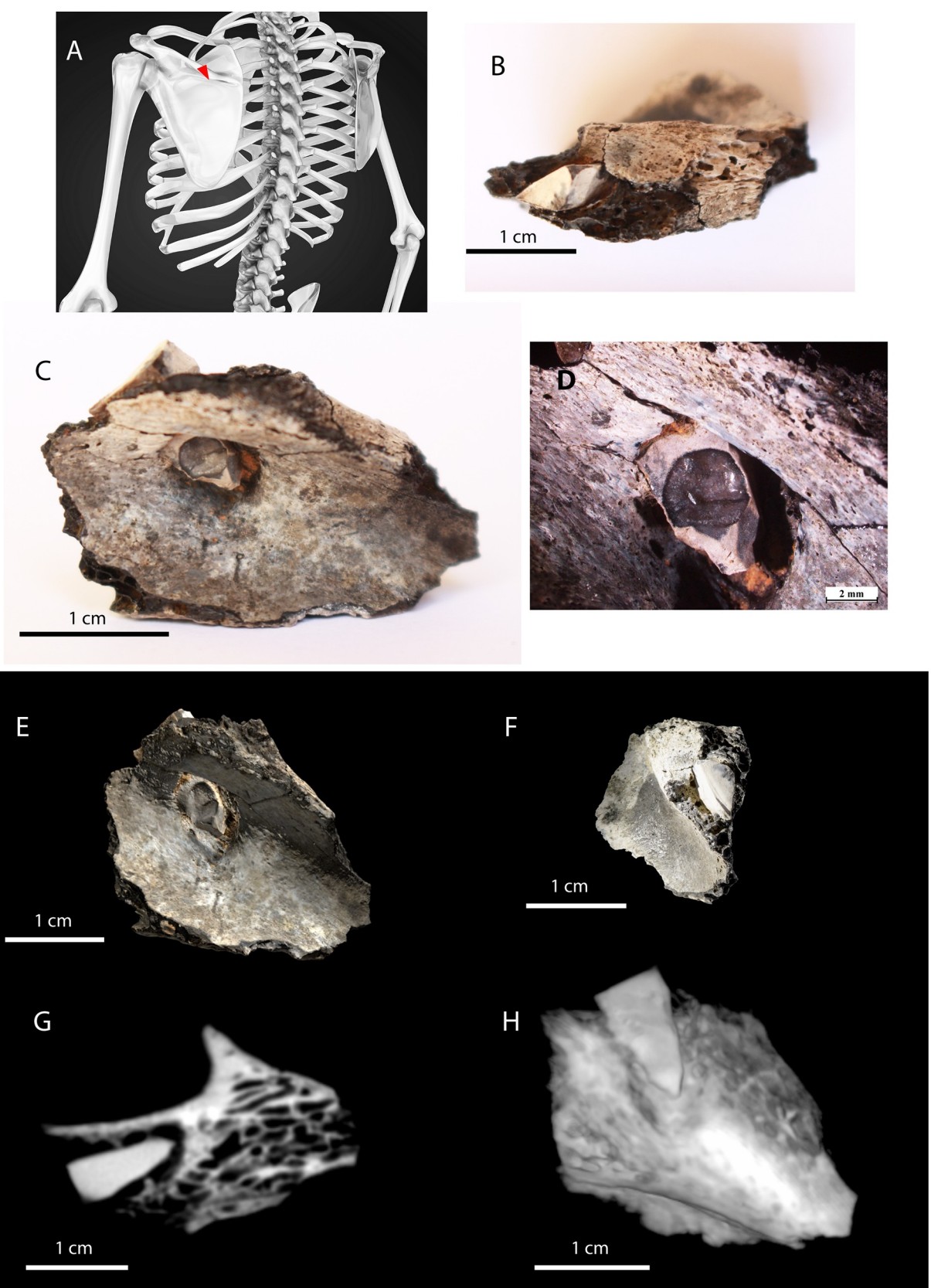

**Fig 5. Projectile point embedded in the left scapula (ref 3209–2).** A: Model seen from its back showing the location and direction of entrance of the projectile point within the bone (red triangle; copyright: Alexmit Can Stock Photo Inc). B: Posterior anatomical view of the fragment of preserved scapular spine. C. Inferior-posterior anatomical view of the fragment of scapular spine. Note the healed bone around the projectile tip. D. Detail of the projectile tip (copyright A. Legrand). E. Photogrammetric acquisition of the inferior anatomical view (copyright R. Brageu). F. Detail of the flexion break of the proximal segment of the projectile point (copyright R. Brageu). G and H: 3Dimage: the projectile point embedded in the bone is fully visible; the complete remodelling of the cortical bone is confirmed (copyright F. Edon and D. Beloeil).

right coxal is found diametrically opposite to this coherent group, lying almost complete not far from the north-eastern wall of the pit. Altogether, despite an absence of articulated joints and dispersion of certain elements, the bones of the axial skeleton show some anatomical coherence.

Neither is the appendicular skeleton lying in total disorder: both sides are not totally comingled but distributed preferentially to the north-western area for the left side and south-eastern area for the right side (Fig 7). The left shoulder, upper arm and forearm (scapula, clavicle, humerus and radius) were found clustered to the north. The bones of the right hand are grouped together at the south-east part of the structure, and the proximal and distal phalanx of the thumb were found articulated. Last but not least, four proximal phalanges of the right foot were found next to each other in anatomical position, with the distal part oriented towards the north, under the right coxal (Fig 8C).

Thus, the inventory of bones and their relative position strongly supports the deposit of an articulated corpse and not dislocated bones (i.e. a primary cremation). The non-random distribution of the appendicular skeleton, the trunk and skull bones suggests that the body was oriented along a south-north axis, with the upper body leaning against the south, south-east or south-west wall of the pit. The cremation would have resulted in the trunk falling forward and its possible rotation. If we consider that the four right pedal proximal phalanges are likely close to their initial position, then it would suggest that the corpse was placed with knees flexed, resembling a seated position. This is also compatible with the narrowness of the pit. Although uncommon, seated burials are known in the Levant since the Epipaleolithic [17, 61–63] and throughout the Neolithic [e.g. 5, 64, 65]. Particularly, in the site of Tell Halula seated burials are exclusive and have been shown to be tightly bound, either with semi-rigid or soft fabrics [5] and placed in pits much smaller than Locus 338. However, as discussed below (section 4.3), it cannot be excluded that the Beisamoun corpse had been placed seated on a pallet above the pyre-pit and during, or after cremation, the burnt remains subsided into the pit.

## 4. The combustion process

### 4.1 Colorimetric and fracture data

The association of heating and colour changes to bones is well established [e.g. 25, 27–29, 66]. Visual examination shows that all shades of discolouration from dark brown to white are present on the bones from Locus 338 (Figs 8–10). To the south-east, some bones of the right hand (triquetral, pollical phalanges, 1st and 2nd metacarpals: S1 Table) do not show signs of discoloration i.e. heating. The hand probably tipped over to the periphery of the heat source early on. The dominant colour of other bones is grey-white to white, that is to say they were close to, or at, the stage of calcination indicative of exposure to high temperatures (see FTIR results below). Transverse and longitudinal superficial cracks, as well as deep cracks, are numerous on most of the bones, including thumbnail fractures. Several bones are distorted and have shrunk, signatures of cremation at high temperatures and compatible with the hypothesis of a primary cremation as argued previously (although the distinction between fleshed, green and dry bones from colour or fracturation is debated [e.g. 25, 27, 29, 55, 56, 66–69]. The skull, the

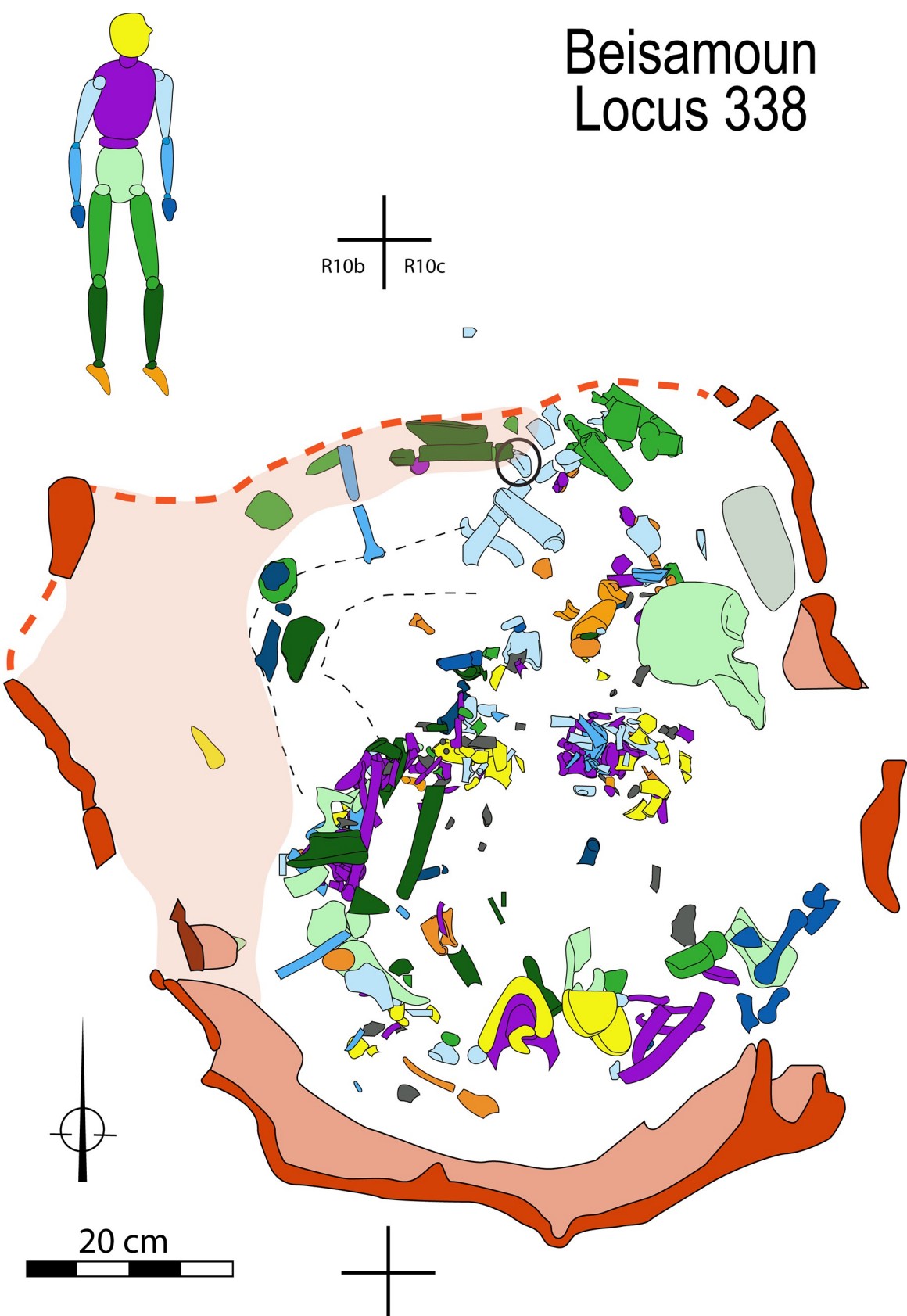

**Fig 6. Spatial distribution of bones within Locus 338 according to anatomical categories.** The injured scapula fragment is circled in black.

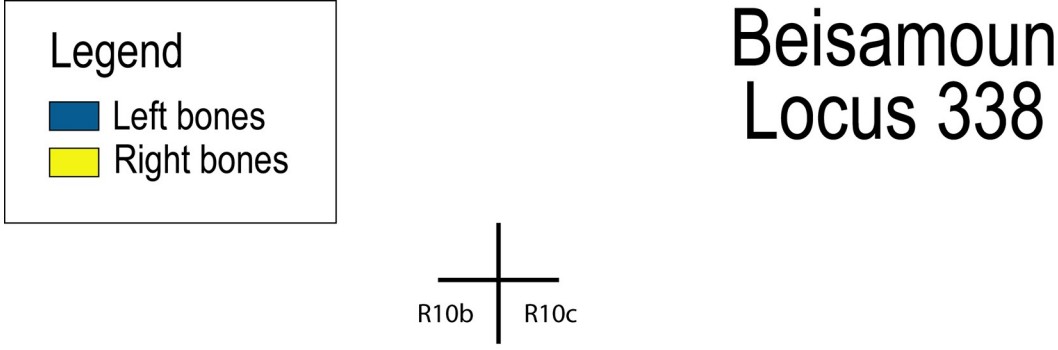

Beisamoun
Locus 338

R10b R10c

20 cm

**Fig 7.** Spatial distribution of right and left sides of the appendicular skeleton.

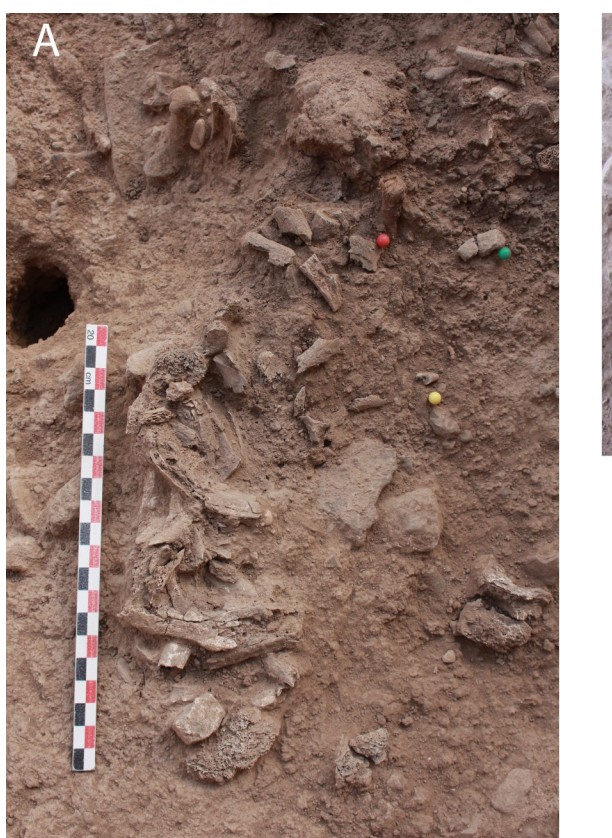
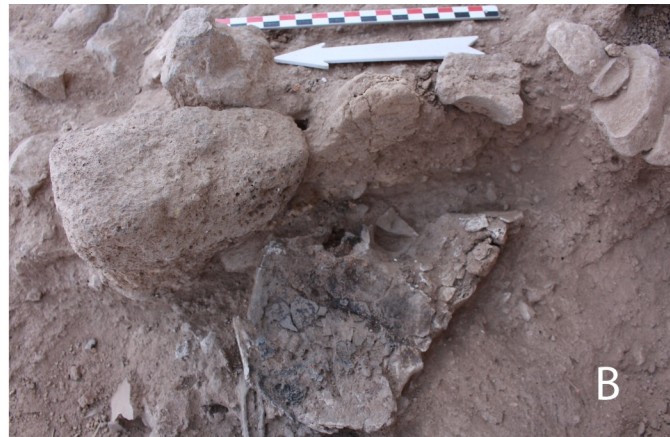
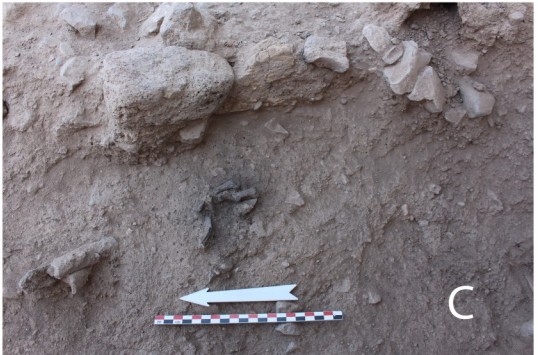

**Fig 8.** Picture of bones *in situ*: A. Segment of axial skeleton: ribs and vertebrae exposed in the middle of the structure. B. Right coxal *in situ*; preserved almost complete by a piece of collapsed mud wall (see Fig 2D). C. Four right pedal proximal phalanges found directly under the right coxal.

vertebral column as a whole, and the left coxal are the most intensely burned elements, followed by the upper limbs and then the lower limbs. It is probable that the position of the corpse itself but also the structure of the bones and the distribution of the fuel in the pit, are factors that have influenced this colour gradation [28–30, 70, 71].

## 4.2 FTIR data

The results of the Fourier transform infrared spectrometry (FTIR) analysis of the bones are summarized in Table 1. FTIR revealed that in the analysed bones—two human skull fragments, one tooth and five other unidentified bone fragments, the original biogenic carbonate hydroxyapatite (CHA) has been partially transformed into hydroxyapatite (HA), indicating that these fragments were heated above 500°C. The human skull fragments in particular show extremely reduced FTIR absorptions of carbonate as expected for bones that have been calcined at temperatures of 700–800°C (Fig 11). Several bone fragments of animals that exhibit the typical black and brown discoloration of charring show no FTIR absorptions of HA. This mineralogical characteristic indicates that these bone fragments were exposed to temperatures below 500°C. In fact, charring of bone collagen occurs at temperatures of between ca. 300 and 400°C [73].

FTIR analysis of the fired mud brick material that lined the pit, shows that the clay minerals that normally compose the mud brick material at Beisamoun [34] has been altered and is missing the characteristic FTIR absorptions of Kaolinite and Smectite at 3695, 3620, 3650 and 915

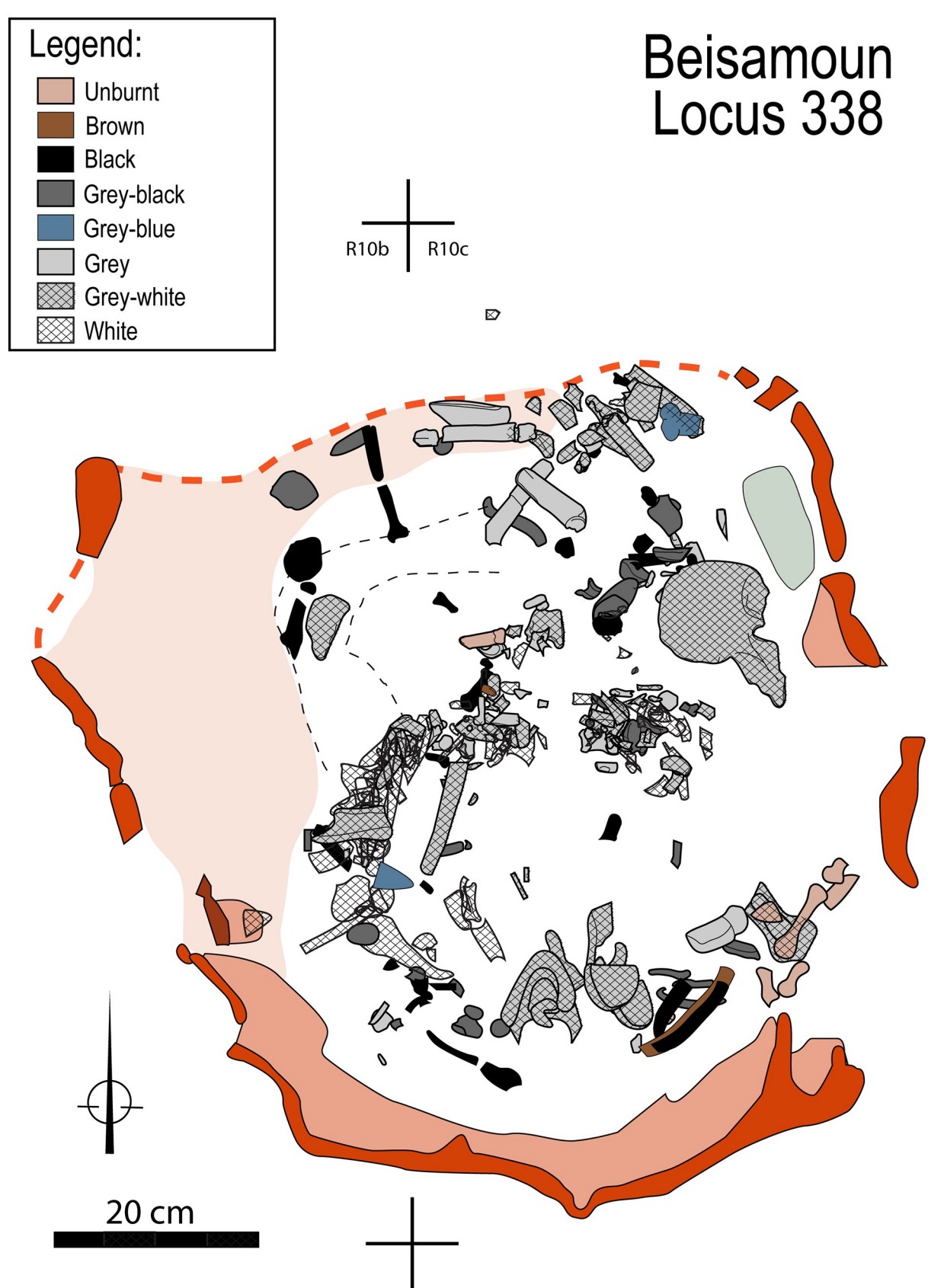

**Fig 9. Spatial distribution of human bones in the pyre-pit according to colorimetric data.**

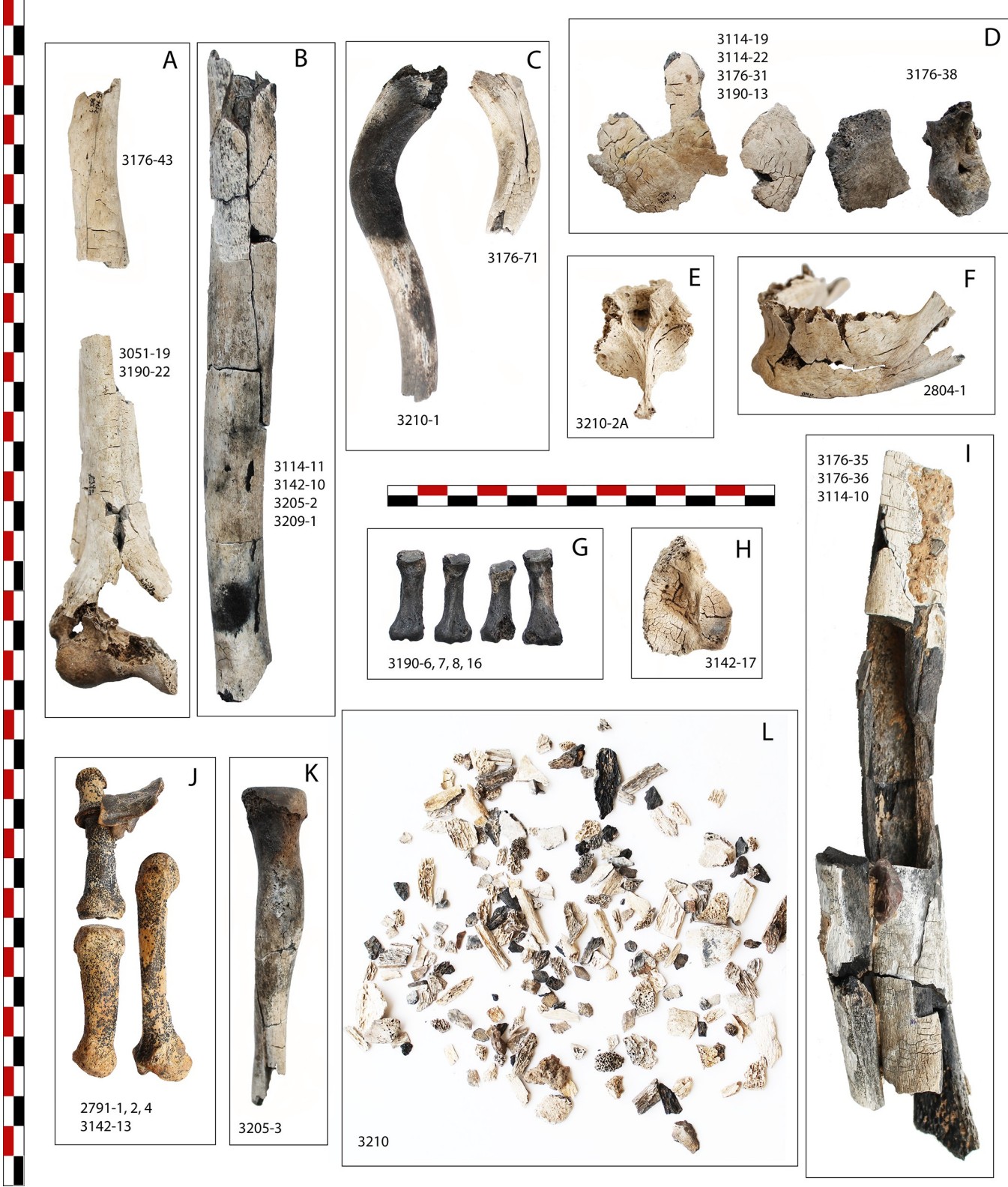

**Fig 10. Photographs of a sample of different types of skeletal elements from Locus 338 after cleaning and refitting process.** A. right humerus; B. left humerus; C. right and left clavicles; D. left parietal and occipital fragments; E. thoracic vertebra; F. mandible; G. right proximal pedal phalanges; H. left calcaneum; I. left femur; J. pollical phalanges, 1st and 2nd metacarpals; K. left radius; L. sieving refuse from spit #3210. For additional information on bone colour and cracks see S1 Table following bone identification numbers.

**Table 1. FTIR analysis of a selection of human and animal bones and teeth and mud brick fragments excavated from Locus 338.** One fragment of cemented wall of the pit, one tooth and 21 bone fragments (fauna and human) retrieved from Locus 338 were analyzed by FTIR to assess the temperature to which they were heated (Table 1, Figs 11 and 12). By using FTIR it is in fact possible to assess if bone and tooth mineral and material containing clay minerals such as kaolinite and smectite were exposed to temperatures above 500° C [72, 73]. Aliquots of a few micrograms of material were scraped from each sample, powdered and mixed with 5mg of KBr. Using a Pike$^{TM}$ hand-press. The mixture was pressed into a 7mm die and analyzed with a Thermo Nicolet iS5 FTIR spectrometer. FTIR spectra were collected by performing 32 scans with a resolution of 4cm$^{-1}$ wavenumbers. The FTIR spectra were analyzed using Omnic software and the SFU Geo-archaeology Laboratory FTIR spectral reference library.

| | Sample ID # | Description | Discoloring | FTIR Analysis* | Integrated interpretation |
|---|---|---|---|---|---|
| 1 | 3102–9 | Humerus prox | Grey | CHA+C+K+Q | heated below 500˚C + soil crust |
| 2 | 3114 | Bone frag | Black | CHA+HA*tr* | heated to ca 500˚C |
| 3 | 3114 | Bone frag | Grey | CHA+C+K+Q | heated below 500˚C + soil crust |
| 4 | 3176–43 | **Human skull frag** | Calcined | HA+CHA | heated above 500˚C below 900˚C |
| 5 | 3210 | **Human skull frag** | Calcined | HA+CHA | heated above 500˚C below 900˚C |
| 6 | 3190 | Bone frag larger | Grey-white | HA+CHA | heated above 500˚C below 900˚C |
| 7 | 3190 | Bone frag smaller | Grey-white | HA+CHA+Q+C+K | heated above 500˚C below 900˚C + soil crust |
| 8 | 3190 | Bone frag | Brown | CHA+C+Q | unheated or heated below 500˚C + soil crust |
| 9 | 3197 | Bone frag | Calcined | HA+CHA | heated above 500˚C below 900˚C |
| 10 | 3197 | Tooth (enamel) | Black | CHA+HA*tr* | heated above 500˚C |
| 10a | 3197 | Tooth (dentin) | Black | CHA | charred below 500˚C |
| 11 | 3197 | Bone frag | Brown | CHA | unheated or heated below 500˚C |
| 12 | 3205 | Bone frag | Unburnt | CHA | unheated |
| 13 | 3209 | Bone frag | Black | CHA+C | charred below 500˚C + calcite crust |
| 14 | 3209 | Bone frag | Brown | CHA | unheated or heated below 500˚C |
| 15 | 3210 | Bone frag | Black | CHA+1617*s* | charred collagen = heated below 500˚C |
| 16 | 3210 | Bone frag | Unburnt | CHA+C+Q | unheated + soil crust |
| 17 | 3142–53 | Tibia shaft frag | Black | CHA | charred and heated below 500˚C |
| 18 | 3142–8 | Bone frag | Unburnt | CHA+C | unheated + calcite crust |
| 19 | 3176–50 | Mandible condyle -subsurface | Brown | CHA | unheated or heated below 500˚C |
| 19a | 3176–50 | Mandible condyle -surface | Brown | CHA | unheated or heated below 500˚C |
| 20 | 3176–77 | Bone frag | Unburnt | CHA+HA*tr*+3570 | heated above 500˚C below 900˚C |
| 21 | 3190–18 | Cattle proximal rib | Unburnt | CHA | unheated |
| 22 | 3 190 | Pig maxilla | Unburnt | CHA+C | unheated + calcite crust |
| 23 | 3176–20 | Cemented mud-brick material | Grey | Q+C+Cl*a*+582 | Mud-brick material heated above 700˚C |

*C = calcite; Cl = clay minerals; CHA = carbonate-hydroxyapatite; HA = hydroxyapatite; Q = quartz; K = kaolinite; *a* = altered; *s* = strong; *tr* = trace.

cm$^{-1}$ wavenumbers (Fig 12). This mineralogical composition supports the hypothesis that the mud brick wall fragment from Locus 338 was heated *in situ* to, or above, 700˚C [72].

In conclusion, the FTIR analysis shows that some of the human bone as well as the mud lining of the pit were heated *in situ* to temperatures well above 500˚C, which is the temperature necessary for corpse cremation. In contrast, most of the faunal remains were less intensely burnt. This result is compatible with the colour and surface modifications (fissures, shrinkage) observed on the human bones identified as calcinated. It is worth noting, on the other hand, the heterogeneous colours on some human remains and the presence of unheated bones. This inter-bone variation may relate to the location of fuel in the pit relative to the corpse and it must be remembered that heterogeneity in bone combustion is expected in the case of corpse cremation [e.g. 25, 28, 70].

## 4.3 Cremation pyre-pit layout

Heat and oxygen flows are major issues in cremation pyre functioning. A crib of wood placed on the ground has been shown to be especially efficient [e.g. 24, 74–76] but leaves very few

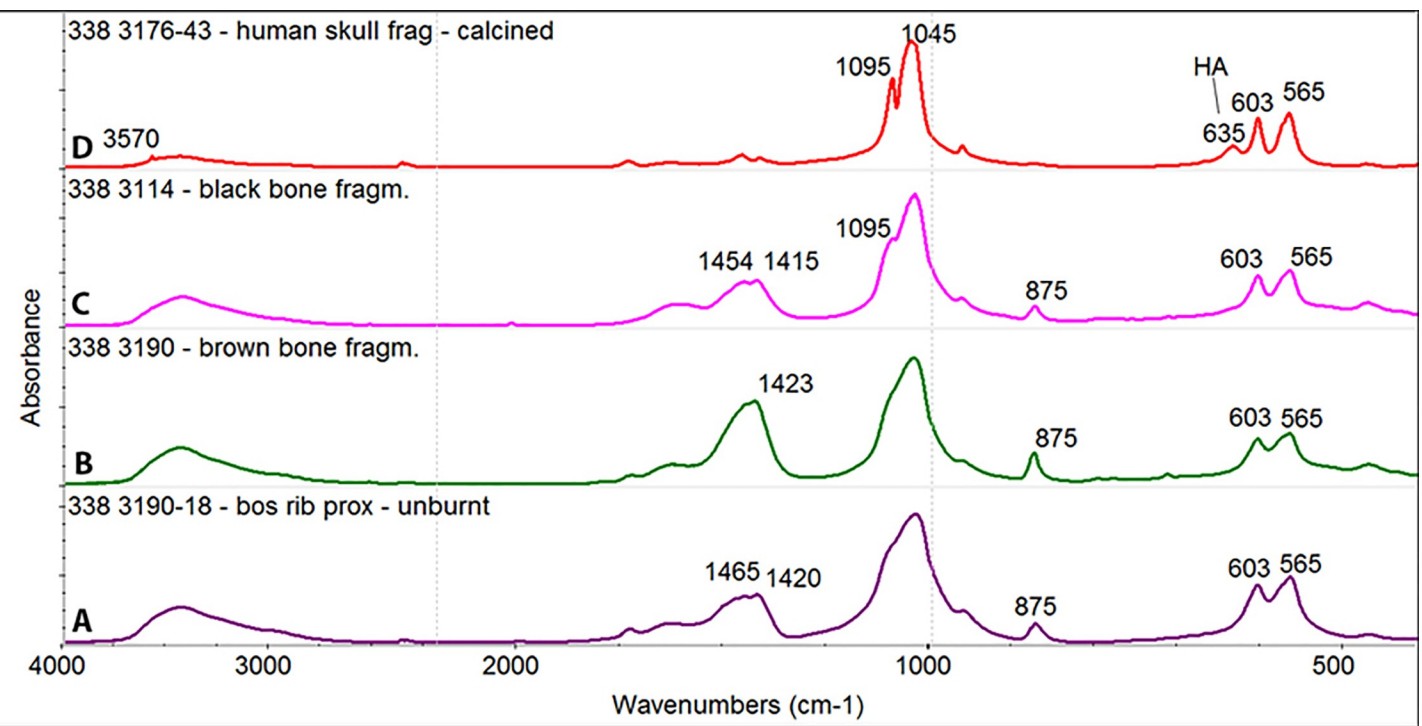

**Fig 11. FTIR spectra of representative bone samples from Beisamoun, Locus 338.** (A) Fragment of cattle rib apparently unburnt to visual inspection. Spectrum shows FTIR absorptions of slightly recrystallized CHA typical of archaeological bone lacking collagen [73]. (B) Bone fragment of undetermined mammal showing brown discoloration. FTIR spectrum similar to (A) with the addition of a strong carbonate absorption at 1423cm$^{-1}$ and weak calcite absorption at 713cm$^{-1}$. (C) Bone fragment of undetermined mammal showing black discoloration. FTIR spectrum similar to (A) with the addition of a weak peak at 1095cm$^{-1}$ suggesting it was heated below 500°C [73]. (D) Fragment of human skull bone showing evidence of being calcined. FTIR spectrum shows HA absorption at 635cm$^{-1}$ and very weak carbonate absorptions at 875 and 1450cm$^{-1}$. This spectral characteristic is typical of bone mineral transformed to HA due to exposure to temperature of between 550°C and 900°C [73].

material traces, which may partly explain why the discovery of archaeological pyres is rare. Cremation pyre-pits are however, attested in Antiquity, for instance the Roman *bustum*, a funerary pyre cum structure. The pit of the *bustum* served as the pyre sub-structure and was left empty precisely for oxygenation of the fire which was placed above it. The corpse lay on top of the wood pile and progressively collapsed, to finally reach the pit sub-structure where it was left and which served as the place of burial [e.g. 71, 77, 78]. Cremation pyre-pits are also known from Bronze and Iron Age periods in Europe, some of them being as small as the Beisamoun structure [54, 79–81]. In the former instances, corpses were placed either within, or on top of, the pit. Based on a great number of archaeological, ethnological and experimental observations, it is worth noting that, even when placed at a high elevation above the fuel, burnt skeletons keep their general anatomical coherence while the corpse and the fuel gradually collapse, all the more so in the case of a untended pyre [24, 28, 54, 74, 79, 82]. Often the corpse is tied up or weighted down in order to avoid movement during cremation [Ibid]. Alternately, corpses can be placed within a pit and this is then covered by fuel such as wood, dung, vegetal chaff and the entire pile covered by a layer of clay. In this case, corpses burn slowly in a reducing atmosphere as observed in Pondicherry, India in the 1990's [28, 74].

At Beisamoun, we observed that the south, east and west walls of Locus 338 show significant rubefaction indicating that the cremation was carried out *in situ*. But similar signs of burning were not visible at the base of the pit. This phenomenon has already been described when the fire is positioned at the base of the pyre, and only reaches a high temperature when it rises in

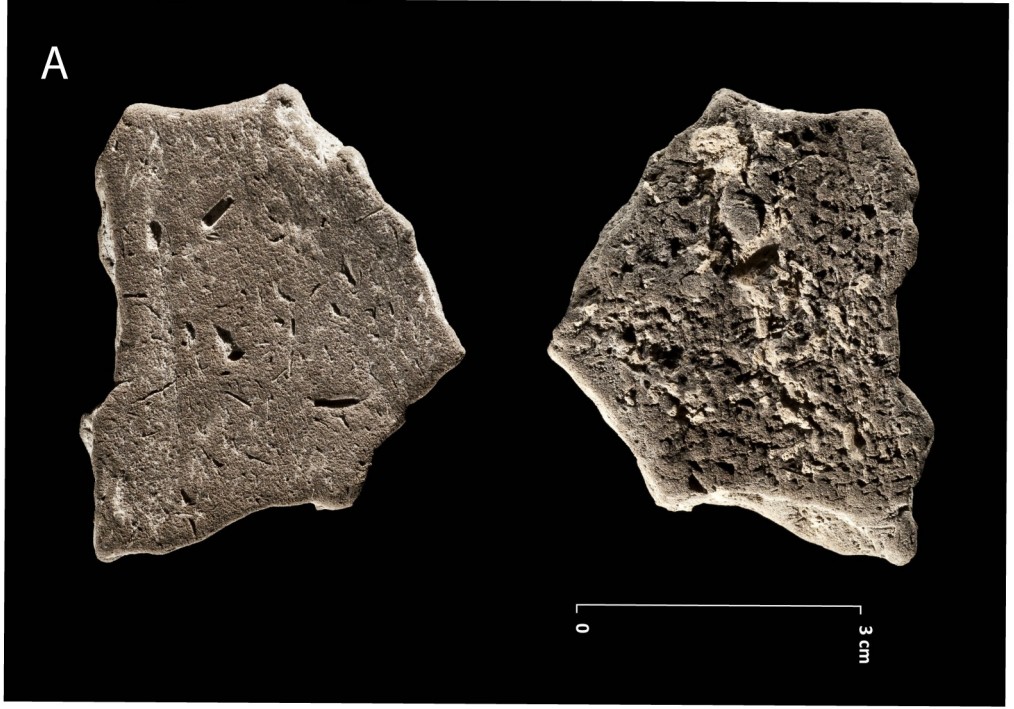

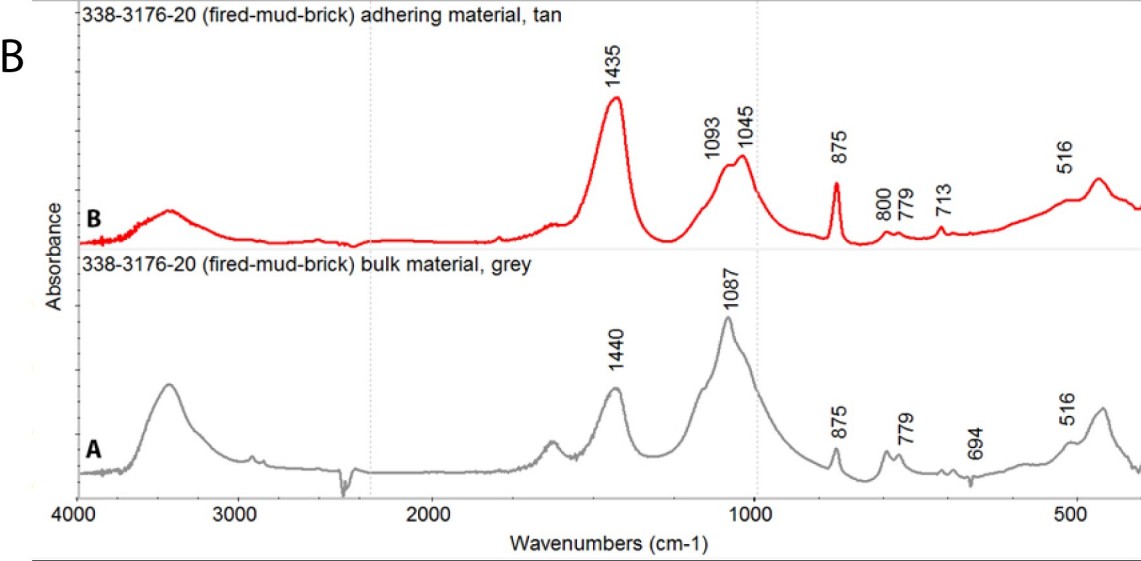

**Fig 12. Pit wall samples from Beisamoun, Locus 338.** A. Picture of the fired mud brick material found in the pyre-pit (see Fig 2) (Ref 3176–20; copyright L. Teira). B. FTIR spectra of sample 3176–20 (A) Greyish brown bulk material. The spectrum shows FTIR absorptions of quartz and calcite and a shoulder at 1050cm-1 assigned to vitrified clay minerals. This type of spectrum is typical of local terra rossa soil heated above 700˚C [72]. (B) Light tan material encrusting the depression of the grass-marks. FTIR spectrum shows the absorptions of quartz (779, 800 cm$^{-1}$), heated smectite (1045cm$^{-1}$ and shoulder at 3620cm$^{-1}$), and calcite (713, 875, 1435 cm$^{-1}$).

an oxidizing environment [e.g. 30, 71, 75, 83]. Therefore, we suggest that the cremation pit functioned like a pit-kiln. It would have been open at the top which would have enabled sufficient oxygen to reach the fire, while the mud walls acted as insulation. Fuel would have primarily been located at the bottom of the pit, but could also have been sprinkled on and about the corpse to augment burning. An empty space where no bones were recovered, located in the

southern half of the pit, could correspond to the heart of the fire around which the most calcined bones were found (Fig 9). The use of a closed pyre as noted in India [28, 74], that was covered, for instance, with a mud plaster-like crust as observed on the wall of the structure, seems unlikely given the dominating light hues of the cremains, rather favouring an oxidizing atmosphere of combustion [e. g. 27, 71].

## 4.4 Fuel

For the combustion to proceed properly, fuel and oxygen must be abundant throughout the cremation process. Body fat and fresh bones can fuel the ongoing combustion, but cannot start it [e.g. 67]. It is essential to use fuel, such as wood or other vegetal matter, to create and maintain a fire, especially one where temperatures over 500˚C are needed. In order to investigate the fuel used for the cremation, phytolith analysis was performed on 10 sediment samples. Eight samples came from the fill of Locus 338 (Samples Bei-14:3–10), and two were control samples from outside the pyre-pit (Samples Bei-14:1–2) (Fig 13). The samples from inside the pyre-pit yielded high densities of phytoliths (measured by weight percent of phytoliths per total sample) as compared to the control samples from outside the structure (Table 2). We also found larger quantities of wood ash ('silica aggregates') than in the control samples (Fig 14).

Another notable plant category identified by phytoliths was sedge (*Cyperaceae*) (stems and leaves). Sedges are water-loving rushes that have spongy stems and are commonly used for basketry and matting. Although we found some sedges in the control samples, there was a notable peak in Sample Bei-14:8, suggesting the possibility that a sedge 'shroud' may have been used to wrap the corpse. The practice of wrapping corpses has been identified as far back as the Natufian period in this region and was quite common in Neolithic contexts all over the Levant [e.g. 5, 16, 84, 85].

In addition to sedges, we also found abundant concentrations of phytoliths from reed grasses in general and *Phragmites* (common reed) in particular. These tall reeds grow along streambeds and around the shores of lakes and wetlands, which characterized the setting of Beisamoun on the Hula Lake. Reed stalks are strong and can be used for construction materials such as for fences, huts, roofing, as well as pallets for beds. The abundant remains of these reeds in the samples from the pit could indicate that *Phragmites* stalks were used as additional fuel, but we cannot rule out the possibility that they might have been used as a shroud or as a pallet on which to rest the corpse above the fire source.

In addition to this evidence for possible sources of fuel, and possible methods for wrapping and treating the dead body before the cremation, we found that the florets of grasses were positively correlated with the stems of grasses within the pyre, indicating the likelihood that grasses were placed into the pit as whole plants. This seems to differ from the more separate distribution of plant parts in the control sample, although the control sample number is too small to be definitive. The grasses in the pyre could have been used as tinder for lighting the fire, or as some sort of ornamentation [86], or as an olfactory screen against the smell of cremation, a practice which is very common in traditional cremation ceremonies today and in the past [78, 87]. In either case, one of the important implications of the presence of grass florets, is the likelihood that the cremation was conducted during the time of year that the grasses were flowering, for example in the late winter or spring.

Finally, we found significant numbers of phytoliths from the husks of wheat, with the highest peak in Bei-14:6, indicating that grains of wheat were present in the pyre, but also in the surrounding area. This could indicate that burial rituals, included the placing of foodstuffs in the cremation pit, although it is also possible that cereals were part of the general sediment matrix or a component of dung used as fuel.

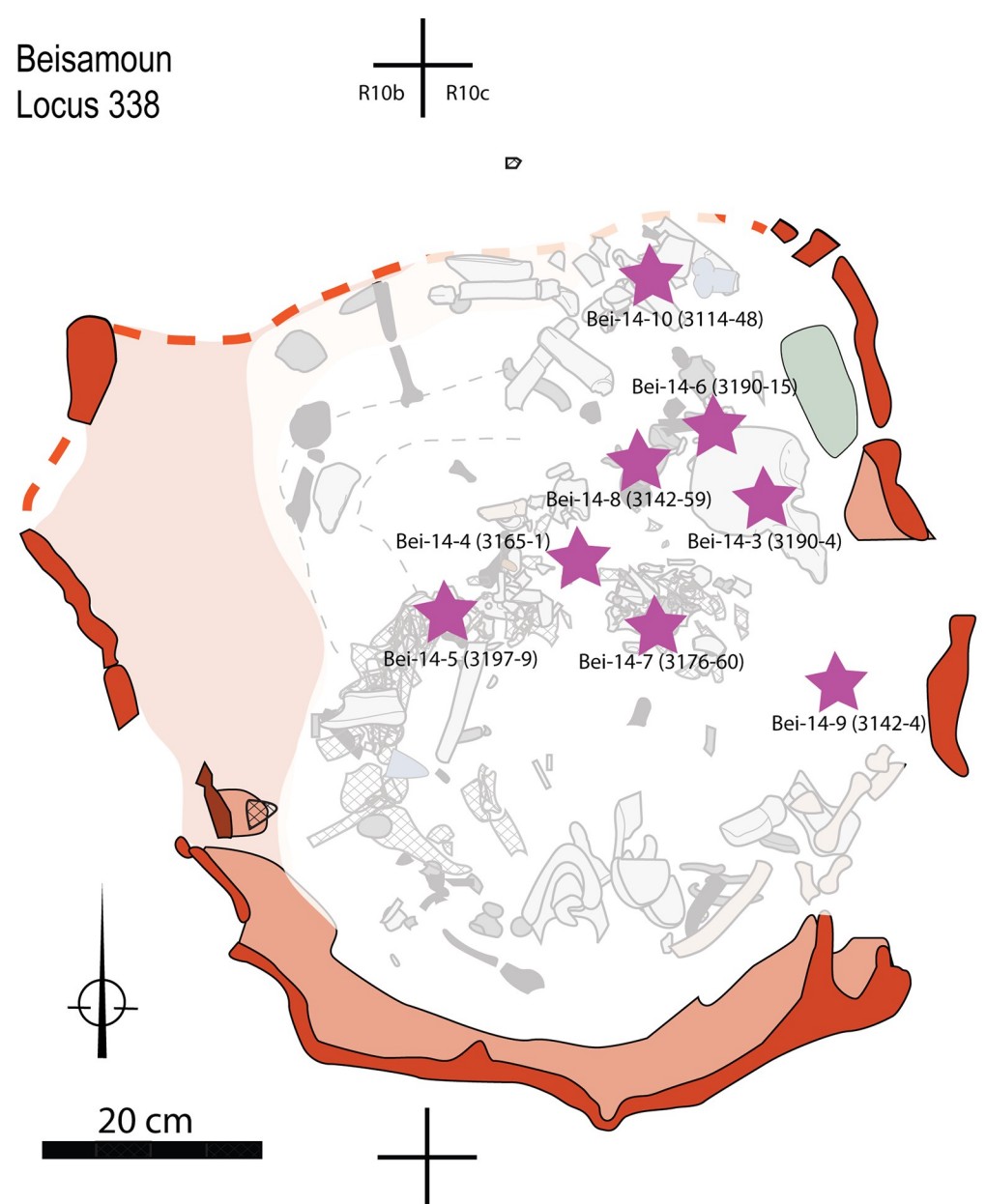

**Fig 13. Spatial distribution of phytolith samples and context list.**

| Lab No. | Square or Locus | Sample | Context | Heights |
|---|---|---|---|---|
| **Bei-14-1** | R10a | 2885-1 | patch of harder sediment | -45/-47 |
| **Bei-14-2** | S9b | 3028-1 | burnt sediment | -43/-45 |
| **Bei-14-3** | Locus 338 | 3190-4 | under complete coxae #3176-32 | -78 |
| **Bei-14-4** | Locus 338 | 3165-1 | block of ashes in the middle of the pit | -76 |
| **Bei-14-5** | Locus 338 | 3197-9 | burnt sediment found at the bottom of the cremation pit | -86 |
| **Bei-14-6** | Locus 338 | 3190-15 | under 4 foot phalanges in anatomical connexion #3190-6,7,8,16 | -78 |
| **Bei-14-7** | Locus 338 | 3176-60 | under bones #3176_43,45,54 | -75 |
| **Bei-14-8** | Locus 338 | 3142-59 | in contact with bone #3142-58 | -76 |
| **Bei-14-9** | Locus 338 | 3142-4 | ashy sediment, next to eastern wall of the pit and bone #3142-40 | -75 |
| **Bei-14-10** | Locus 338 | 3114-48 | north of pit and under bones #3114-8,10,11 | -77 |

**Table 2. Raw phytolith counts in numbers per gram sediment.** The sediments were first separated into aliquots of ca. 800 mg of sediment in the < 250 micrometer size fraction. These were treated with 10% HCl to remove pedogenic carbonates. We then added a dispersant of NaP to disaggregate clay particles. The fine sands and silts were settled out of the samples by allowing the material to fall through a column of water, after which we decanted the clay particles. The remaining sand and silt was dried, then burned in a muffle furnace at 500˚C to remove organic matter. The sediment was then floated in a heavy density liquid which allowed the phytolith particles to float. Phytoliths were decanted, washed and mounted onto microscope slides. At least 300 phytoliths were counted for each sample.

| Sample: | Bei-14-1 | Bei-14-2 | Bei-14-3 | Bei-14-4 | Bei-14-5 | Bei-14-6 | Bei-14-7 | Bei-14-8 | Bei-14-9 | Bei-14-10 |
|---|---|---|---|---|---|---|---|---|---|---|
| **SINGLE-CELL** | **n /gm** | **n /gm** | **n /gm** | **n /gm** | **n /gm** | **n /gm** | **n /gm** | **n /gm** | **n /gm** | **n /gm** |
| Long-Cell (Psilate) | 72473 | 101699 | 103346 | 138738 | 129430 | 87024 | 163947 | 257644 | 149205 | 140270 |
| Long (Echinate) | 18423 | 18734 | 55501 | 54731 | 62431 | 56368 | 77566 | 143461 | 70214 | 71729 |
| Long (Sinuate) | 3687 | 2676 | 6698 | 3818 | 1523 | 11867 | 22917 | 17567 | 8777 | 9564 |
| Long-Cell (Cylindroid) | 8595 | 3747 | 6698 | 8910 | 18272 | 16811 | 33495 | 32205 | 14628 | 3188 |
| Long-Cell (Dendritic) | 125303 | 25157 | 71768 | 96734 | 147702 | 75157 | 135741 | 184449 | 105321 | 98826 |
| Papillae: | 8595 | 1606 | 11483 | 1273 | 7614 | 5933 | 14103 | 17567 | 14628 | 4782 |
| Stoma | 1234 | 1071 | 0 | 1273 | 1523 | 0 | 0 | 0 | 0 | 0 |
| Hairs: | 7375 | 4282 | 1914 | 7637 | 15227 | 13845 | 14103 | 20494 | 11702 | 3188 |
| Trichomes: | 11062 | 5353 | 3828 | 10183 | 6091 | 8900 | 3526 | 8783 | 7314 | 6376 |
| Bulliform: | 33173 | 21410 | 14354 | 33093 | 21318 | 25712 | 35257 | 114183 | 27793 | 23910 |
| Fan-shaped Bulliform | 8595 | 6423 | 2871 | 14001 | 12182 | 7911 | 15866 | 52700 | 19016 | 12752 |
| Crenates | 12282 | 3212 | 10526 | 6364 | 6091 | 3956 | 14103 | 11711 | 11702 | 6376 |
| Polylobes | 0 | 0 | 1914 | 0 | 0 | 0 | 0 | 0 | 1463 | 3188 |
| Bilobes: | 1234 | 1606 | 9569 | 0 | 7614 | 3956 | 8814 | 0 | 10240 | 3188 |
| Rondels: | 25798 | 4817 | 39233 | 12728 | 57863 | 20767 | 42309 | 46844 | 33644 | 63759 |
| Saddles: | 11062 | 1071 | 12440 | 8910 | 12182 | 6922 | 8814 | 17567 | 8777 | 4782 |
| Cones: | 3687 | 0 | 0 | 2546 | 3045 | 3956 | 1763 | 0 | 1463 | 7970 |
| Horned Tower: | 1234 | 0 | 0 | 0 | 4568 | 1978 | 1763 | 0 | 0 | 0 |
| Flat tower | 0 | 1071 | 0 | 0 | 0 | 1978 | 1763 | 5856 | 0 | 0 |
| Elongate psilate | 1234 | 0 | 0 | 0 | 0 | 0 | 0 | 0 | 0 | 0 |
| Echinate Spheroid | 0 | 0 | 0 | 0 | 0 | 0 | 0 | 0 | 0 | 0 |
| Globular psilate Spheroid | 0 | 0 | 957 | 1273 | 1523 | 0 | 0 | 0 | 0 | 0 |
| Sclereids | 0 | 535 | 0 | 0 | 0 | 0 | 0 | 0 | 0 | 0 |
| Tracheids | 0 | 535 | 957 | 1273 | 0 | 989 | 0 | 2928 | 1463 | 1594 |
| Blocks | 3687 | 0 | 0 | 2546 | 1523 | 0 | 0 | 2928 | 0 | 0 |
| Platelet | 1228 | 1071 | 3828 | 8910 | 7614 | 0 | 19392 | 11711 | 4388 | 12752 |
| Single Polyhedron | 0 | 1071 | 957 | 1273 | 3045 | 1978 | 0 | 0 | 2926 | 0 |
| Compound platelet | 0 | 535 | 0 | 0 | 0 | 2967 | 0 | 0 | 0 | 1594 |
| Silica aggregate | 18423 | 5888 | 19138 | 61095 | 35022 | 4945 | 15866 | 158099 | 71677 | 47819 |
| Coarse Verrucate | 0 | 0 | 0 | 0 | 1523 | 0 | 0 | 0 | 0 | 1594 |
| **MULTI-CELL** | **n per gm** | **n per gm** | **n per gm** | **n per gm** | **n per gm** | **n per gm** | **n per gm** | **n per gm** | **n per gm** | **n per gm** |
| Leaf/Stem: | 0 | 0 | 11483 | 7239 | 0 | 0 | 2350 | 1403 | 0 | 0 |
| LS Psilate | 15304 | 6423 | 0 | 0 | 20699 | 19036 | 23505 | 26655 | 22277 | 17306 |
| LS Sinuate | 665 | 1071 | 478 | 0 | 1428 | 2596 | 1175 | 7014 | 1310 | 0 |
| LS Echinate | 4658 | 2676 | 957 | 0 | 4996 | 0 | 0 | 5612 | 1310 | 2472 |
| LS Stomata | 0 | 0 | 0 | 0 | 0 | 2596 | 3526 | 0 | 0 | 0 |
| LS Saddle | 0 | 0 | 0 | 0 | 714 | 0 | 0 | 0 | 1310 | 0 |
| Panicoid leaf/stem | 1331 | 3747 | 1435 | 2758 | 3569 | 1731 | 0 | 1403 | 1310 | 3708 |
| Unident Husk: | 1996 | 5353 | 1914 | 1379 | 6424 | 13845 | 9402 | 11223 | 10483 | 8653 |
| Wheat Husk | 1996 | 1606 | 957 | 689 | 2141 | 3461 | 2350 | 1403 | 0 | 0 |
| Barley Husk | 665 | 0 | 0 | 345 | 0 | 0 | 3526 | 0 | 0 | 3708 |
| Aegilops | 665 | 0 | 0 | 0 | 0 | 0 | 0 | 0 | 0 | 0 |
| Wild Grass Husk: | 13308 | 6423 | 4306 | 3103 | 15703 | 9518 | 15278 | 11223 | 5242 | 4945 |

(*Continued*)

**Table 2.** (*Continued*)

| Sample: | Bei-14-1 | Bei-14-2 | Bei-14-3 | Bei-14-4 | Bei-14-5 | Bei-14-6 | Bei-14-7 | Bei-14-8 | Bei-14-9 | Bei-14-10 |
|---|---|---|---|---|---|---|---|---|---|---|
| Cyperus C- cones | 3327 | 5888 | 2392 | 1379 | 4283 | 6922 | 4701 | 5612 | 2621 | 4945 |
| Phragmites Stem | 1331 | 2141 | 478 | 689 | 714 | 1731 | 3526 | 0 | 1310 | 1236 |
| Juncus type | 0 | 0 | 0 | 0 | 0 | 0 | 0 | 0 | 0 | 0 |
| Panicoid leaf/stem | 1331 | 3747 | 1435 | 2758 | 3569 | 1731 | 0 | 1403 | 1310 | 3708 |
| Bromus-type Stem | 0 | 0 | 0 | 689 | 0 | 0 | 0 | 1403 | 0 | 0 |
| Avena | 0 | 0 | 0 | 0 | 0 | 0 | 0 | 0 | 0 | 0 |
| Setaria-Type Husk | 0 | 535 | 478 | 0 | 0 | 0 | 8227 | 2806 | 9173 | 3708 |
| Phragmites Leaf | 3992 | 3212 | 1435 | 689 | 1428 | 1731 | 7051 | 0 | 3931 | 3708 |
| Cyperaceae leaf/stem | 13308 | 5888 | 4785 | 1724 | 5710 | 3461 | 8227 | 11223 | 5242 | 2472 |
| Sedge stem | 0 | 535 | 0 | 0 | 0 | 0 | 0 | 0 | 0 | 2472 |
| Reed grass stem | 0 | 0 | 0 | 689 | 714 | 0 | 1175 | 0 | 1310 | 0 |
| Cereal Straw | 0 | 1606 | 0 | 0 | 1428 | 865 | 1175 | 1403 | 0 | 0 |
| Scirpus-type | 0 | 0 | 0 | 0 | 0 | 865 | 0 | 1403 | 0 | 0 |
| Multi-tiered forms | 665 | 0 | 0 | 0 | 0 | 0 | 0 | 0 | 0 | 0 |
| Polyhedron | 2662 | 0 | 0 | 345 | 1428 | 3461 | 0 | 0 | 5242 | 2472 |
| Polyhedral hair base | 0 | 0 | 0 | 0 | 0 | 865 | 0 | 0 | 0 | 0 |
| Verrucate | 0 | 535 | 0 | 0 | 0 | 0 | 0 | 0 | 0 | 0 |
| Coarse Verrucate | 0 | 0 | 0 | 0 | 714 | 1731 | 0 | 1403 | 0 | 0 |
| Awn | 665 | 0 | 957 | 689 | 714 | 0 | 1175 | 0 | 0 | 1236 |
| Cyperus sp. | 0 | 0 | 0 | 0 | 714 | 0 | 0 | 0 | 0 | 0 |
| Leaf/stem with irregular ronel/cone | 0 | 0 | 0 | 0 | 0 | 0 | 2350 | 0 | 0 | 0 |
| Leaf/stem jigsaw | 0 | 0 | 0 | 0 | 0 | 0 | 1175 | 0 | 0 | 0 |
| Square-cell leaf/stem | 0 | 0 | 0 | 0 | 0 | 0 | 0 | 0 | 1310 | 0 |
| Melted phytolith | 0 | 0 | 0 | 689 | 0 | 0 | 0 | 0 | 0 | 0 |
| Low conf. palm leaf | 0 | 0 | 0 | 345 | 0 | 0 | 0 | 0 | 0 | 0 |
| Low conf. wheat | 665 | 535 | 0 | 0 | 0 | 0 | 0 | 0 | 0 | 0 |
| Multiple square cells | 0 | 0 | 957 | 0 | 714 | 1731 | 0 | 0 | 0 | 0 |
| High conf. emmer | 665 | 0 | 0 | 0 | 0 | 0 | 0 | 0 | 0 | 0 |
| Total Phytolith Wt % | 3.56 | 3.97 | 3.67 | 5.82 | 6.39 | 3.43 | 4.68 | 4.91 | 8.27 | 8.34 |

## 4.5 Pyre tending and manipulation of cremains

While part of the human bone assemblage is highly fragmented, which is not surprising in a cremation, it should be noted that several bones retained their integrity or almost (right coxal, mandible—despite the fracturing of all teeth—metatarsals), and that the diaphyses of the long bones were relatively unfragmented. This may relate to the corpse having been exposed to very limited combustion conduction with minimal technical intervention, leaving it to burn at a slow rate, untended [e.g. 24, 28, 31 54, 74, 79, 82]. It could also explain the significant heterogeneity of combustion temperatures revealed by bone colour and FTIR investigation. It should be noted that, even without external intervention, under the action of fire, corpses are susceptible to trunk rotation and sometimes substantial limb movements due to muscle contraction following exposure to heat [e.g. 28–30, 70, 71]. Subsequently, the gradual subsidence of the wood pyre (and burning of the fabric binding) may have facilitated the dislocation of the skeleton. As they fell, the bones then spread around the source of the fire, further disrupting their initial position and orientation.

There is no evidence for the intentional manipulation of the corpse during or after cremation. However, pre-cremation damage was observed in the form of five striae on the

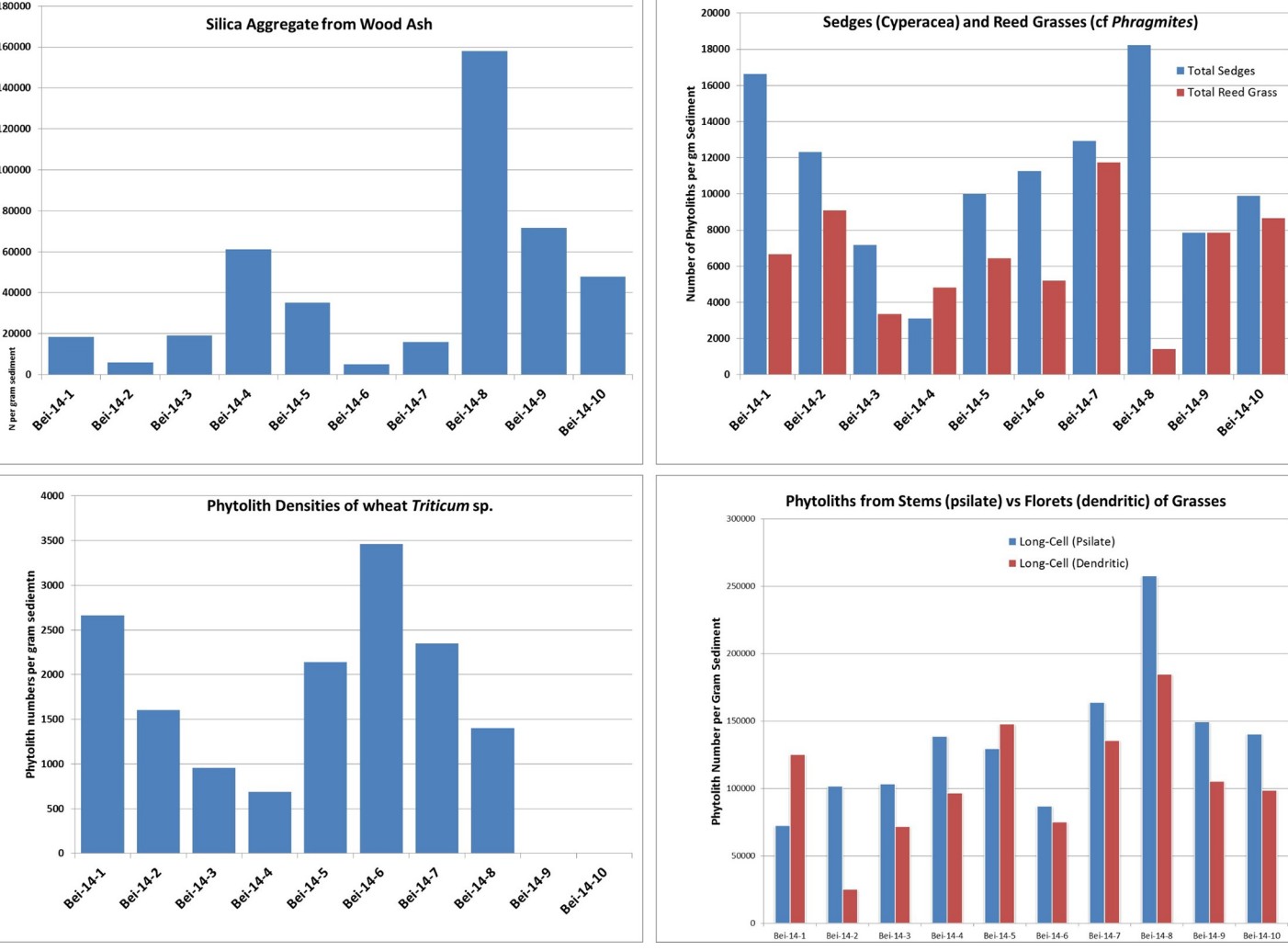

**Fig 14. Diagrams of phytoliths outside (samples 14:1 and 14:2) and inside the pyre-pit (all other samples).** A: Graph showing densities of 'silica aggregates' from the ash of wood from trees and shrubs. B: Graph showing concentrations of sedge and reed grass (cf. *Phragmites*) phytoliths. C: Graph showing concentrations of phytoliths from wheat. D: Phytolith densities of grass stems and grass florets indicating flowering grasses were a component of the burial.

posterolateral face of the posterior talar facet of the right calcaneus (the talus, on its side was too fragmentary to be examined). These striae are wide, the longest are also the shallowest. The morphology of these striae indicate the action of a carnivore rather than human intervention (Fig 15). In addition, on the edge with the *sustentacular sulcus* there is a small blunt and bright area, as if polished. The rest of the collection was meticulously observed a second time looking for additional surface traces on other bones, but without success. Although there is an animal burrow at the bottom of the pyre pit, it belongs to a rodent and not a carnivore. Therefore, the observed modification on the calcaneus should be considered as the result of an isolated event that took place prior to cremation. A possible explanation could be damage to the foot by a carnivore (such as a dog) as the corpse lay prior to being cremated. The articulated phalanges of the right thumb found in the pyre suggests a short delay between death and cremation.

In conclusion, the data collected and presented here, strongly support the contention that the Locus 338 corpse was cremated whole, as a fresh corpse. The burnt mud pit lining, the pit fill as well as the total bone weight, skeletal element representation, the temperature-related

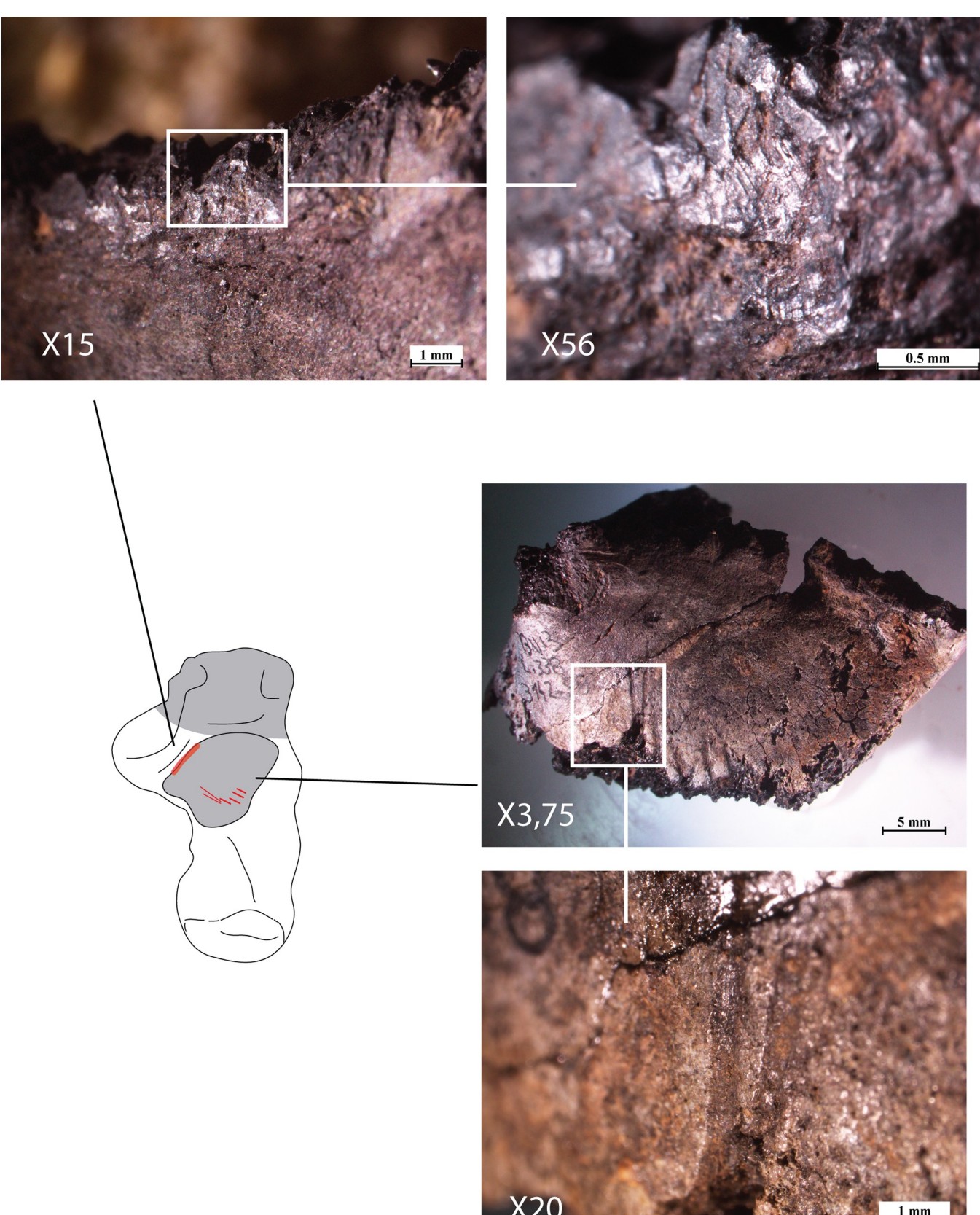

**Fig 15. Location of the striae and polish area observed on the right calcaneus.** Copyright A. Legrand.

alterations evident on the bones, together with their spatial positioning within the pit, support the interpretation of the cremation as having been that of a complete, fresh corpse rather than isolated dry bones. Despite this, we cannot totally rule out the possibility that the corpse was cremated after a delay during which it desiccated. Such delayed burials are suspected at Neolithic Çatal Höyük [12] but are as yet unknown at Beisamoun despite a detailed archaeothanatological analysis of all skeletons recovered. Indeed, all the primary burials at the site show a regular pattern of decay *in situ* and follow the expected steps of joint disarticulation, which is the opposite of delayed burial [e.g. 88]. Neither is their position at burial excessively flexed, as can be observed in mummies. Even with exposure outdoors in an arid environment, a long period of time is needed after death to achieve dehydration and reduction of desiccated tissue [89, 90]. Exposure of skeletal elements and their bleaching and exfoliation may follow. At Beisamoun no signs of the latter have been observed on any of the skeletons, mediating against the idea that they were exposed for any length of time following death. Moreover, the location of the site in the Mediterranean phytogeographic zone, adjacent to the swamps of the Hula Lake in the Jordan Valley that is extremely humid year-round (mean temperature is 16˚C in January and 33˚C in August; monthly average relative humidity is 71% December to 53% in June; average annual precipitation 550mm [91]), is an unsuitable environment for dessication, which requires high temperature and low humidity such as found in the caves in the Judean desert [92]. This is why, if the hypothesis of desiccation cannot be wholly excluded, it also cannot be entirely favoured. The only way to equivocally support the idea of cremation of a desiccated corpse would be to find histological evidence of bacterial attack of the bones typical of advanced corpse decay prior to cremation [e.g. 93, 94].

## 5. The pit fill

Only one artefact was present in the pit, a fragment of a spatula made of mammalian bone. Its fragmentary state does not favour an intentional deposit but the question remains open. A total of 776 other faunal remains were recovered from the cremation pit. The majority (NISP = 692) represent unidentified splinters of mammalian bone and teeth retrieved during sieving of the deposit. Only 84 remains could be identified to species and/or skeletal element. These are listed in Table 3 and represent at least eight taxa or faunal body size classes including common Levantine wild and domestic ungulates, birds of prey and freshwater fish.

Table 3. Faunal taxa represented in pyre-pit Locus 338.

| Species | NISP |
|---|---|
| Cattle (*Bos* sp.) | 7 |
| Caprine (*Capra/Ovis* sp.) | 2 |
| Goat (*Capra* sp.) | 2 |
| Gazelle (*Gazella* sp.) | 1 |
| Pig (*Sus* sp.) | 13 |
| Large Mammal | 5 |
| Medium Mammal | 39 |
| Small Mammal | 2 |
| Raptor (Bird of prey) | 1 |
| Fish (Pisces sp.) | 7 |
| Cyprinids (e.g. Carp) | 2 |
| Cichlids (e.g. Tilapia) | 3 |
| **Total identified** | **84** |
| **Total unidentified frags** | 692 |
| **Total Fauna** | 776 |

The question is whether the fauna recovered were intentionally placed in the pyre-pit or are accidentally associated with it (Fig 16). If intentional then they may have served as fuel for the cremation or represent grave offerings that were incinerated with the deceased. Alternately, they were simply part of the sedimentary matrix of the underlying Layer Ic into which the cremation pit was dug, and so were accidentally burnt. Species identified in the pit as well as skeletal element representation and unidentified bone fragment size, mirror those found in adjacent squares as well as in other parts of the site [33–35]. Within Locus 338, a wide range of body parts are represented with no discernible element selection or spatial patterning (Table 4). Table 5 and Fig 17 present the data on colouration of all faunal remains recovered from the pit and illustrates the broad spectrum of colours. Burnt animal remains may be the result of intentional burning resulting from cooking, refuse removal through burning in a fire, use of bone as fuel or due to intentional cremation [e.g. 95–101]. However, bones may also be burnt due to unintentional actions (e.g. post-depositionally), as demonstrated elsewhere [102–105]. These studies have shown that bones lying underneath or adjacent to hearths or on cooled ashes may be altered in colour and texture, though apparently less so than bones that were in direct contact with a fire. Many methods are in use to determine burning. Here we have chosen three approaches- scoring colour [66, 103], bone size and type [e.g. 106–108] microscopic examination (x70) of the bone surface for alterations such as cracking [after 69] and FTIR analysis [109] to assess burning temperature. Several experiments [e.g. 68, 102, 110] have shown that when burnt, dry bones tend to have lighter hue (brown shades) compared to fresh bones, and exhibit slightly less surface damage (fissures) or shrinkage. Consequently, the degree of burning (colouration) may reflect the state of the bones when exposed to combustion.

It is evident that the majority of remains show some degree of burning While, 31% of the remains are unburnt or only singed (classes unburnt to brown/black in colour). A total of 22% are grey to white which is similar colouration to the calcined human remains. Moreover, there was no patterning to the burning based on whether the skeletal element was an unidentified splinter or a diagnostic bone. In terms of their spatial arrangement, there are slightly more unburnt bones in the southern part of the pit than in the northern part, with a concentration of blackened bones towards the centre of the pit, which follows the patterning for the human remains. Compared to quantities of burnt remains in squares outside the pit, significantly more burnt remains were found within Locus 338 and a higher number were calcined.

It has been argued [e.g. 96, 108] that high proportions of small sized (<2 cm) burned fragments of spongy bone reflect their use as fuel. Most bones in the Beisamoun pit (NISP = 636) represent very small fragments- both cortical and spongy bone- under 10mm in length, while 144 bones had lengths over 10mm (Fig 18A). Even so, the majority of bones are small and fall in the 10 to 30mm length range, with very few over 60mm in length. Likewise for breadth the majority of bones were under 5mm in width, while the larger remains peaked at 10 to 15mm (Fig 18B). As demonstrated in Table 6, there is a correlation between bone size (length, breadth, depth measurements) and colour, with the burnt bones e.g. ranging from black to grey, overall having smaller dimensions. Since burning makes bones more friable and so more susceptible to fragmentation [103, 107], this may account for the observed patterning rather than their use as fuel.

Experiments on modern bone used as fuel demonstrate that spongy bones burn better than dry bone or compact bone without the cancellous portion [95, 96, 107, 108, 111]. Thus, if bone was used for fuel, spongy bones should dominate the burnt assemblage. Examination of the relative frequency of cortical shafts to trabecular-rich bones (spongy bones) in the pyre-pit (Table 4), shows that only 34.5% represent spongy bones (ribs, vertebrae, horncore). Moreover, as demonstrated above, the vast majority of the faunal remains exhibit a far lower degree

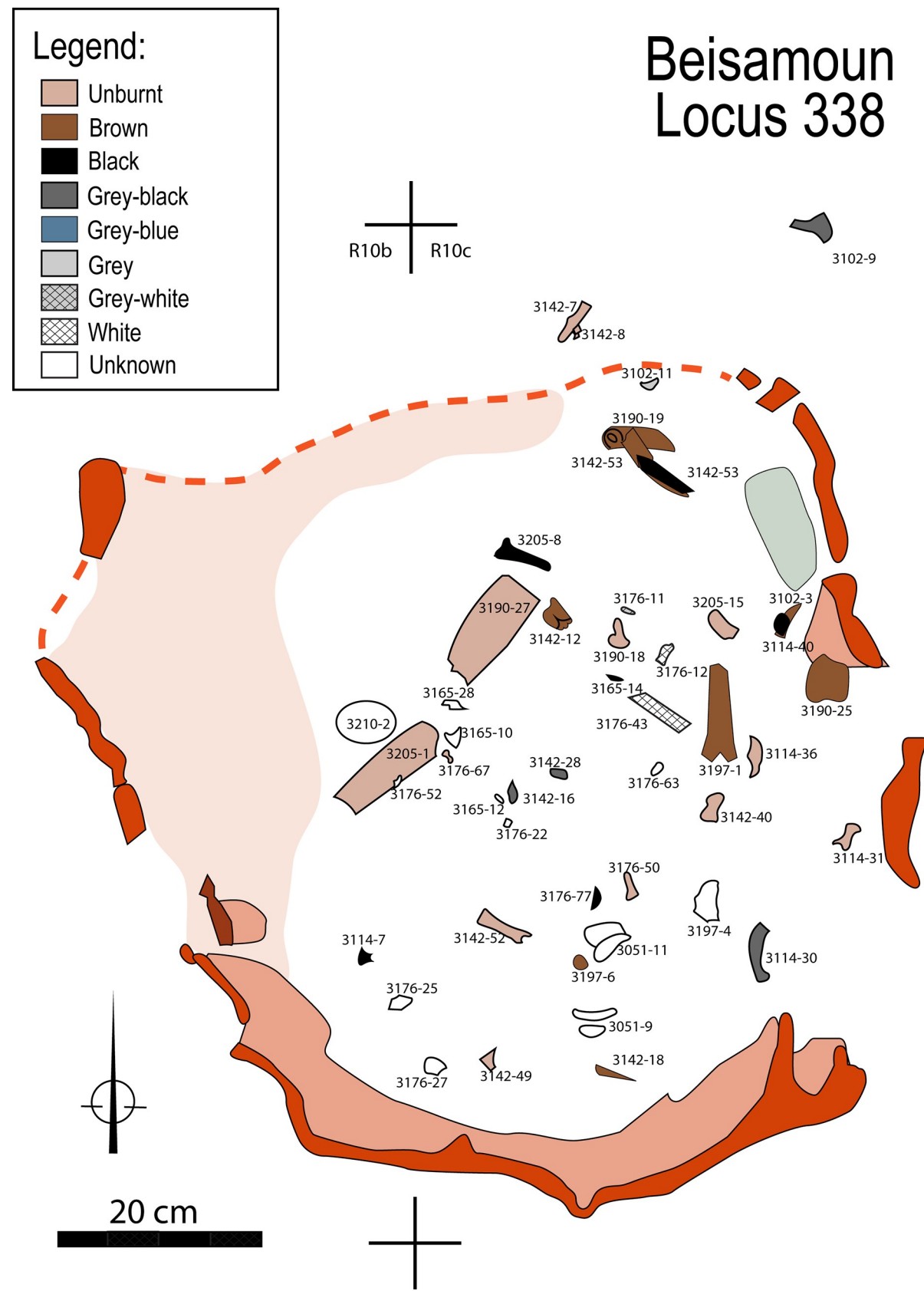

**Fig 16. Spatial distribution of faunal remains from the pit according to colorimetric data.**

**Table 4. Faunal skeletal element representation.**

| Element | NISP |
|---|---:|
| horncore | 1 |
| cranial | 8 |
| mandible | 2 |
| teeth | 10 |
| rib | 11 |
| Vertebra (includes 12 fish vertebrae) | 17 |
| scapula | 3 |
| humerus | 2 |
| radius | 1 |
| metacarpal | 1 |
| pelvis | 1 |
| femur | 1 |
| tibia | 2 |
| longbone shafts | 15 |
| calcaneum | 1 |
| carpals/tarsals | 2 |
| 1st ph | 4 |
| 2nd ph | 2 |
| **Total NISP** | **84** |

of burning than the calcined and cremated human remains. Consequently, it is highly unlikely that the bones found in the pit served as fuel for the cremation supported by the low proportion of spongy bone in the assemblage as a whole, the presence of unburned spongy bone and the presence of burnt teeth that do are unsuitable as fuel.

From the pit, all identified bones and a sample of 50 unidentified fragments were subjected to a microscopic examination of the surface texture at x70 magnification. Cracks and exfoliation on the cortical surface were only observed on the calcined bones. Pérez et al. [105] noted that fresh bone exhibits extensive changes to the cortical surface (in color, a high degree of fragmentation, cracks and exfoliation = structural changes on the cortical surface), while dry bone shows only changes in color. If, the bones in the pit were dry when burnt, this may explain the absence of alterations to the bone surface as well as the more minor burning relative to the human remains.

Non-anthropogenic modifications were observed on only two bones; small carnivore puncture holes were evident on a humerus shaft and on an unidentified large mammal shaft

**Table 5. Colour of faunal remains.**

| Colours | NISP |
|---|---:|
| Unburnt | 91 |
| Brown | 107 |
| Brown-Black | 42 |
| Black | 178 |
| Black-Grey | 183 |
| Grey-White | 74 |
| White | 101 |
| **Total NISP** | **776** |

**Fig 17. Pie chart showing breakdown of burning on faunal remains in the cremation pit.**

fragment (Fig 19A). In addition, a percussion fracture was evident on the same humerus shaft as noted above (Fig 19B) and another shaft fragment exhibited a flaked end. This random representation of damage to the bones further supports the faunal assemblage as part of the background fill of the pit.

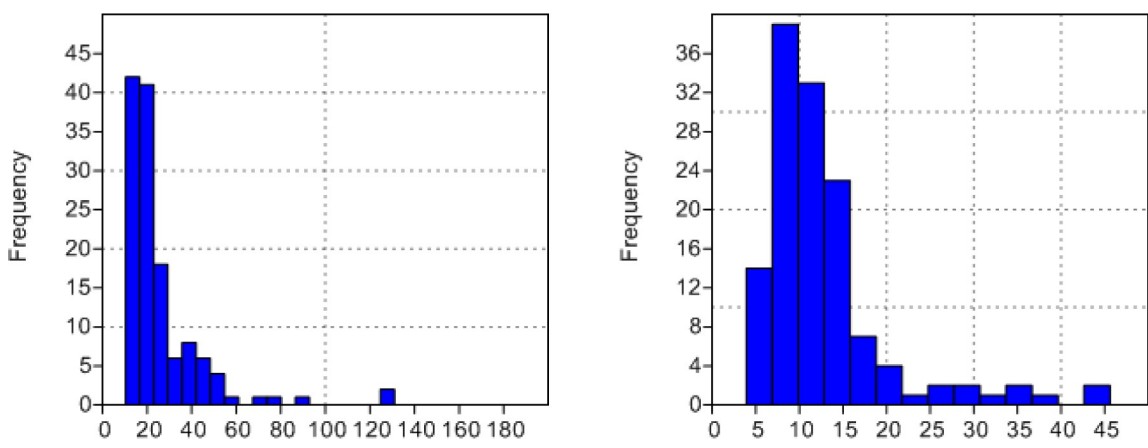

**Fig 18.** Histograms of bone size (a) length (b) Breadth. Missing are bones <10mm in length and <5mm in breadth.

**Table 6. Colour-size correlation.**

| | | NISP | L | B | D |
|---|---|---|---|---|---|
| Unburnt | X | 40 | 31.78 | 15.05 | 10.44 |
| | max | | 131.01 | 45.61 | 123.6 |
| | min | | 11.51 | 5.63 | 2.42 |
| Brown | X | 21 | 31.82 | 14.91 | 8.40 |
| | max | | 86.74 | 42.87 | 45.26 |
| | min | | 12.27 | 5.56 | 2.31 |
| Brown-Black | X | 20 | 21.82 | 12.19 | 6.60 |
| | max | | 37.08 | 19.82 | 12.52 |
| | min | | 12.39 | 4 | 2.68 |
| Black | X | 27 | 20.86 | 10.48 | 5.73 |
| | max | | 51.17 | 24.82 | 17.82 |
| | min | | 10.28 | 4.67 | 2.47 |
| Black-Grey | X | 16 | 20.82 | 9.23 | 6.13 |
| | max | | 43.85 | 13.96 | 10.34 |
| | min | | 10.62 | 3.91 | 2.98 |
| Grey-White | X | 15 | 19.62 | 10.44 | 6.48 |
| | max | | 24.33 | 15.78 | 19.8 |
| | min | | 12.03 | 6.95 | 2.64 |

To conclude, the data presented here on the animal remains from Locus 338 all point to their representing accidental inclusions, remains that were already present in the Layer Ic sediment into which the pit was dug. This is based on several factors: (i) the lack of evidence for selection of specific taxa and/or skeletal elements since a wide spectrum of both are represented, and all taxa are found in other parts of the site; (ii) the absence of any discernible spatial structure in the location of the faunal remains within the pit–they are dispersed throughout; (iii) the small size of most of the remains–with the majority representing unidentified splinters less than 10 mm in dimension- suggesting a refuse deposit; (iv) the uneven pattern of burning -ranging from unburnt to calcined–and relative paucity of bones burnt to the same degree as the human remains–reflect the proximity of the remains to the cremation. The range of colours and the low frequency of spongy parts, also negates the use of bones as fuel; (v) the low frequency of structural changes to the cortical bone surface on most bones, suggests that the majority were already dry when burnt-which may also account for the lower degree of burning observed. (vi) The presence of disparate taphonomic modifications on the bones including carnivore and human damage, indicate different agents of modification and so origins. Consequently, the animal remains do not present a cohesive assemblage and as such do not form an integral part of the cremation or rites associated with it. (vii) Finally, the fact that the fauna from the pit resembles those recovered from adjacent squares in terms of species and skeletal element representation as well as fragment size, but differs in terms of the quantity of burnt bones and their colour, supports the identification of the pit fauna as simply elements incorporated in the sediment matrix of the pyre-pit.

## 6. Abandonment of the pyre

About 15 to 20 cm of the upper northern half of the pit is eroded but it is entirely preserved at its southern end. The lower two-thirds are filled with ashy sediment, in which the human remains were found (Fig 20). Above the pit, in the upper 20 to 25 cm, the sediment is identical to the surrounding matrix (representing Layer Ib). Some segments of the pit wall collapsed

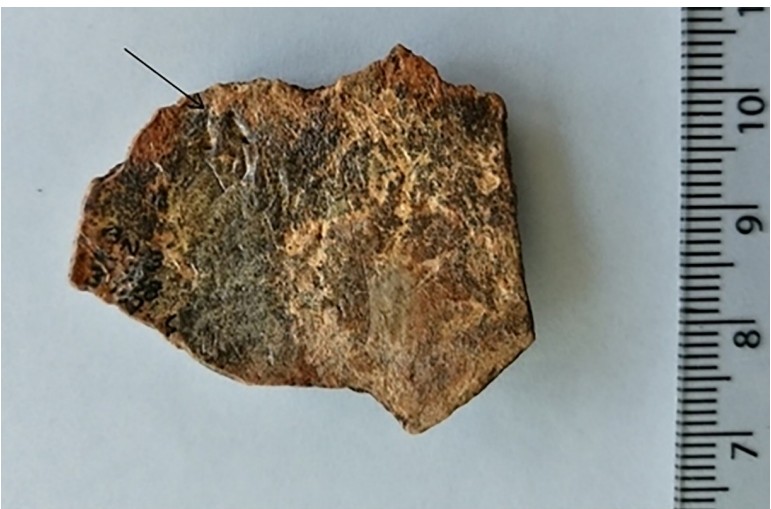

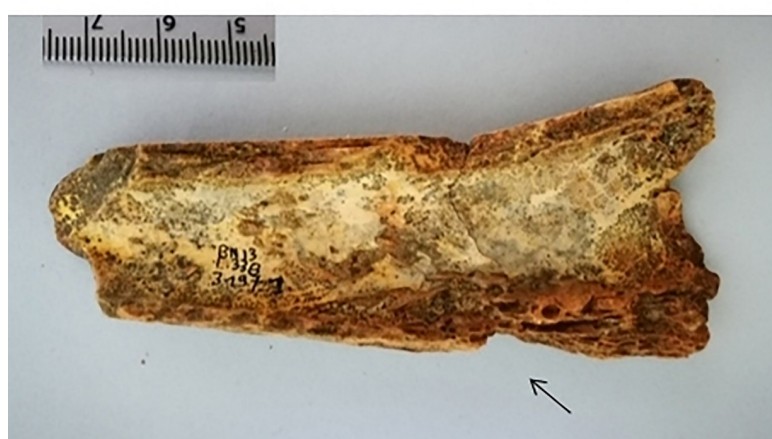

**Fig 19. Non-anthropogenic modifications observed on faunal assemblage.** (a) Large mammal bone fragment with carnivore tooth pits. (b) Large mammal humerus shaft with a percussion fracture.

inside the structure, sealing the layer of ash and bone (Fig 2). This suggests that once the cremation was completed, the structure was left abandoned as it was and remained open. This hypothesis is reinforced by microstratigraphic analyses carried out on several samples of the wall, which show that its upper part has been strongly altered by weathering. However, the removal of bones after cooling cannot be excluded (which may partly explain why the skeleton is incomplete). In addition, the presence in other localities at the site of secondary deposits of burnt bones, attests to a tradition of selective bone collection destined to be buried elsewhere (see below). In the case of Locus 338, it is difficult to identify which bones were removed intentionally, which were destroyed by burning or which were removed by bioturbation. However, relatively few remains are missing as attested by the total weight of the cremated bones and skeletal element representation.

## 7. Other cremations at Beisamoun

To date, aside from Locus 338, five other cremations are known from the site (two in Sector F and three others in Sector E). Amongst them three were excavated and two were found in the last excavation season, documented and left *in situ*: Loci 246 and 347 were fully excavated;

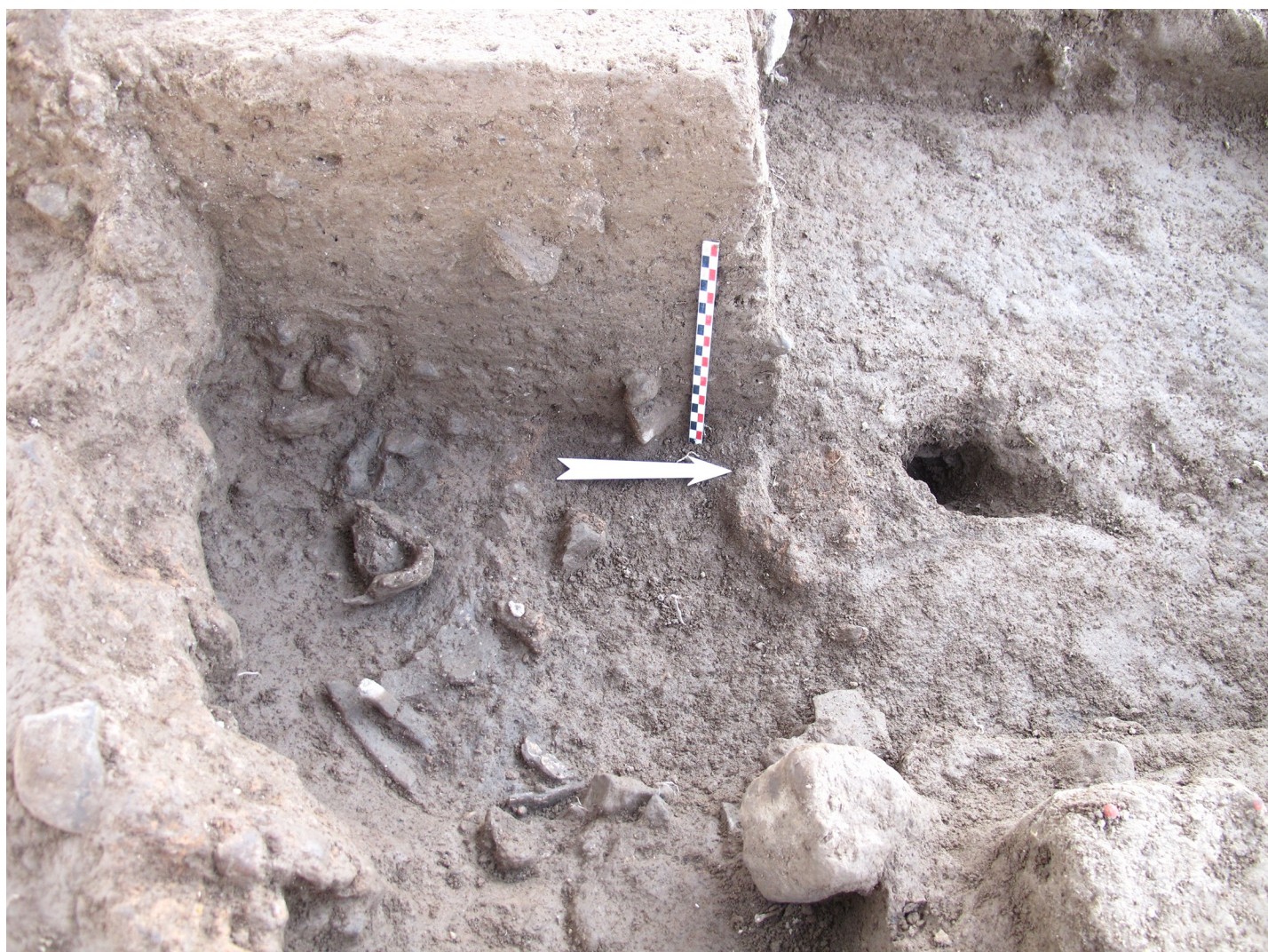

**Fig 20. West section of the south-east fourth of the pit. ongoing excavation.** Note the ashy sediment at bottom sealed by Layer Ib matrix. Hole on the right: burrow.

Locus 435 and 460 were partly excavated; Locus 211 was partly destroyed by a mechanical shovel. Unlike Locus 338, all represent assemblages of selected bones that had been collected together with some ash from a pyre. None occur in deep, mud lined pits, as evidenced in Locus 338. Their respective inventories show that all categories of bones were selected but in small quantities (Fig 21). For each such assemblage the MNI estimate is one adult and colouration and damage on the bones suggests that they originated in a primary cremation of a fresh corpse (breakage, cracking, shrinkage). The bones of Locus 347 delimited a partial circle, attesting that they were likely deposited in a container, possibly a basket [34: Fig 32]. In Locus 211 a worked, perforated and burnt limestone cylinder was deposited with the cremated human remains [34: Fig 25].

The calcinated femur of the adult from Locus 211 (Upper Layer of sector F) was directly dated to 7084–6831 cal BC (Beta-519962; 8060±30 BP; 97.7% of confidence BetaCal3.21), placing it contemporaneous with the pit-pyre Locus 338. According to our stratigraphic observations, the three other secondary cremations should be contemporaneous or slightly earlier to

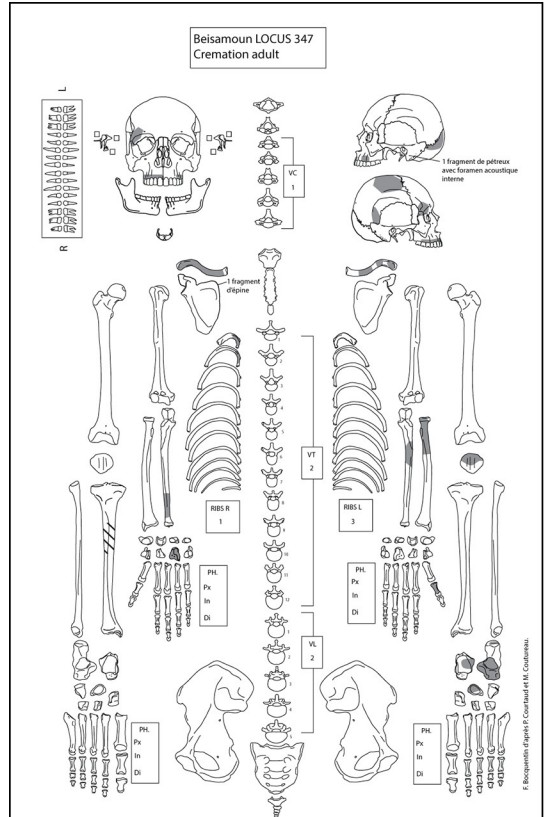

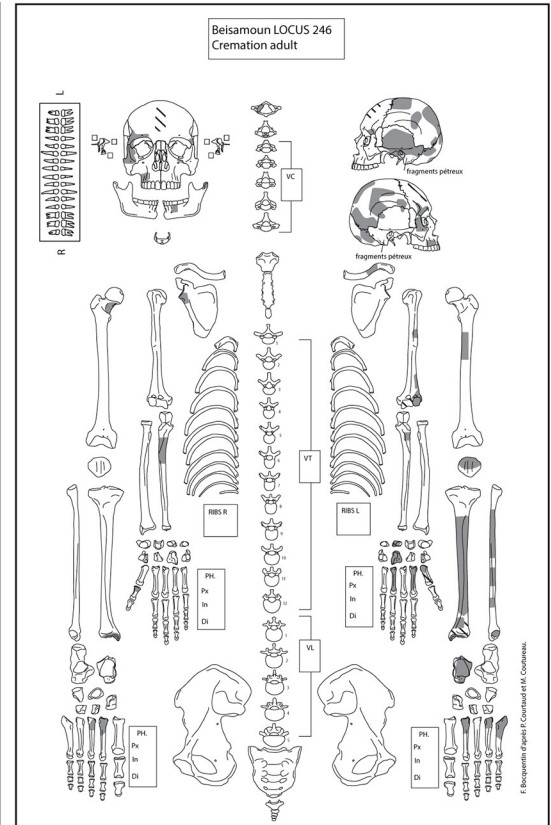

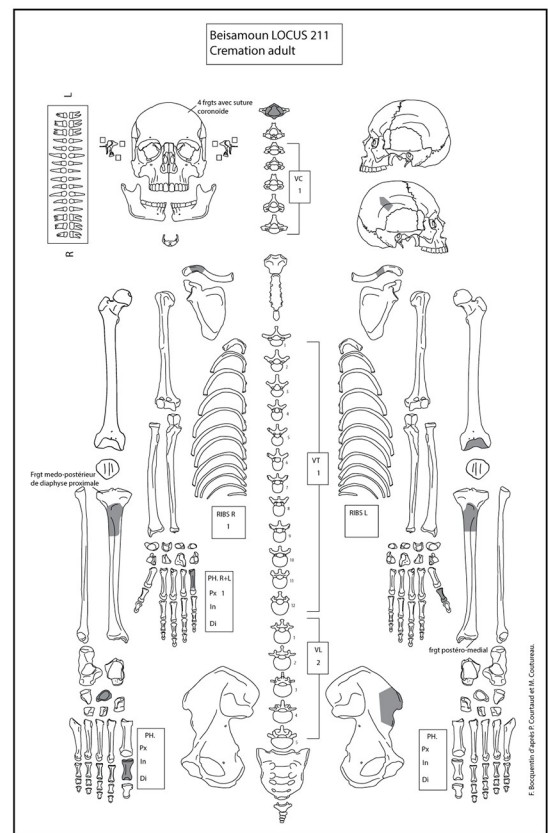

**Fig 21. Inventory sheets of the fully excavated, three secondary deposits of selected cremated remains from Beisamoun (Loci 211, 246, 347).** The MNI is one adult respectively.

these dated examples. These cremations are all contemporaneous with other types of funerary treatment of adults at the site. In contrast, immature individuals were buried as primary deposits throughout the entire occupational sequence.

## 8. Discussion

Locus 338 at Beisamoun is a cremation pyre-pit that served as well as the final repository for the cremains. What remained of the deceased (based on bone weight, skeletal element representation, and element spatial distribution, pattern of burning on the bones), the evidence for *in situ* pyrotechnic activities (ashy pit fill, burnt mud plaster on the pit wall) and the associated finds (phytoliths, fauna) allows us to confirm that it represents a single cremation event of a fully articulated corpse.

The bones and the pit exhibit signs of combustion at a high temperature (suited to burn a fresh corpse) resulting in typical damage to fresh bone (distortion, shrinkage, splitting, longitudinal cracking). Overall, some bones have retained their anatomical coherence, including labile joints (feet phalanges; ribs/vertebrae; shoulder/arm left). According to other pyre-pit structures in the literature and published experimental work, the body may have been placed either within, or above, the pit which was filled with combustibles and left open. It would have gradually collapsed into the pit as the fuel was consumed. Whether initially inside or above the pit, it's the lower limbs were likely to have been bound in a flexed position. The body had its back towards the south or south-west wall of the structure, either as a result of having been initially placed in the pit in this position, or else took up this stance once it subsided into the pit. The cremated human bones and ash were left in place and later enclosed by the collapse of the burnt pit wall.

It should be noted that although corpse cremation is attested to in approximately contemporaneous Mesolithic European contexts [112 and references therein] and in Neolithic Arabia [113], the scarcity of evidence for pyres means that our understanding of the associated combustion processes, is as yet, very partial for these Early Holocene cremations.

The findings from Beisamoun demonstrate that cremation treatment first appeared at the site with the transition from the 8[th] to the 7[th] millennia BC i.e. the very beginning of the FPPNB/PPNC, a cultural transitional still poorly understood. Immediately before, in the PPNB deposits at the site, adults are found in primary or secondary deposition and attest to skull removal and plastering (two badly damaged plastered skulls were found in the 2016 excavation season [37]). With the appearance of cremation in the PPNC, primary and secondary burial at Beisamoun continued, but skull removal and plastering seem to disappear. By the end of the PPNC, inhumation and cremation both disappear and there are only a few primary burials of children interred on-site before its definitive abandonment.

To date, the primary cremations at Beisamoun and the pit-pyre in Locus 338, are possibly the most ancient of the Near East. However other cases of intentional fire induced modification have been documented in the region. Grave 556 at Tell Aswad (Syria) attests to the intentional firing of dry bones in a Middle PPNB context (8200–7500 cal BC). The remains of a young child appear to have been selectively removed from a primary burial, redeposited and burnt *in situ* as the pit shows clear traces of burning [8: 212–213]. The bones, that are beige to black in colour, appear to have been exposed to a moderate temperature. Very recently, a case of charred remains gathered in a secondary multiple burial, and directly dated to 7058–6825 cal BC, i.e. contemporaneous to the Beisamoun cremations, was published from the site of

Kharaysin (northwest Jordan) [22, 23]. The comprehensive geochemical and spectroscopic data indicate moderate burning temperatures applied to already skeletonized bones. Thus, the analysis supports an intentional secondary cremation as in Tell Aswad.

In addition, strikingly similar mud plastered pyre-pits to that found at Beisamoun were found at the slightly later site of Tell el-Kherk in the Northern Levant and are attributed to the Middle Pottery Neolithic level dated between 6600 and 6100 cal BC [7, 20, 21]. However, they differed notably in their function: remains of all ages, including infants, were found and most pits were used to inter several deceased, up to ten in number. This quantity of individuals packed in structures not larger (about 1 m diameter) than Locus 338 at Beisamoun, convinced the authors of the Tell el-Kherk publication, that the deceased must have been skeletonized or at least partially decomposed prior to cremation (some joints are still articulated). Moreover, at Tell el-Kherk, mostly skulls and long bones were present indicating some selection of body parts. A detailed anthropological analyses would probably be necessary to arrive at conclusions regarding the *chaîne opératoire* in this case. Nevertheless, the fact that the bones are found almost complete, with transverse breaks and the majority do not seem to be calcinated but only carbonized (from observation of the photos) would indeed support the hypothesis of secondary cremations. Some anthropologists in this case may refute the use of the term « cremation », as no attempt is made here of active defleshing by fire. However, the deliberate treatment by fire of the remains at Tell el-Kherk is obvious considering the pyre structures and the number of skeletons involved. It should be distinguished from instances of accidental burning and all cases where the human remains are burned not as first intention (intentional destruction of the grave structure by fire, proximity to a fire place etc...[see for instance 13]). A few centuries later, during the 6th millennium in Halaf contexts, cremation and pyre-pits are also documented in sites further to the east such as Yarim Tepe II [114] and further to the north-west at Mesrin [115].

In the Southern Levant, charred bones are mentioned in one PPNC grave at Ashkelon and at Atlit Yam, but in both instances are considered as part of secondary disturbances [116, 117]. Indeed, cremation is absent at several dozens of FPPNB/PPNC graves in other Levantine sites located further to the south and west ('Ain Ghazal; Motza). It demonstrates that cremation was not practiced everywhere at this time. This would suggest that cremation emerged in the Central Levant and spread to the Northern Levant where large cemeteries appear in parallel [e.g. 21, 115]. Indeed, the similarity of the cremation pits in the subsequent PN at Tell el-Kherk, makes coincidence unlikely, even if the practice appears to have evolved and changed. It suggests that some elements of Northern-Southern Levantine interaction did continue during the 7th millennium despite the lag in the introduction/development of pottery manufacture in the Southern Levant.

The FPPNB/PPNC cremation activities demonstrate the custom of an alternative funerary program which might have had a strong impact on funeral procedure, mourning time and even ritual meaning. The differences observed in fire-use between the contemporaneous sites of Beisamoun and Kharaysin either for active defleshing in the first case or as secondary manipulation of skeletonized remains in the latter, augurs a diversity of practices within common landmarks–the use of fire—that is fascinating. We cannot exclude that the final result is part of a shared funerary ideology. In general, there are multiple reasons for the practice of cremation that differ from one society to another, and the symbolism it conveys is just as varied and can be positive (privileged) or negative (sacrilege) [e.g. 31, 87, 118, 119]; for the post-Neolithic Near East see e.g.: 120]. Beyond the defleshing process, the desire to fragment and destroy the corpse completely, cannot be excluded. Indeed, colour changes, cracking, fragmenting and distortion of bones induced by heat at high temperature might be part of the funerary intention.

The pyre-pit structure found at Beisamoun, together with the secondary cremains deposits, show a complex operational chain of funerary gesture with strong technical constraints and several steps of handling. This involvement contrasts with other cultural aspects seen at the site which seem to be technically under invested when compared to the previous PPNB (e.g. lime-plaster manufacture, house building, flint knapping). At PPNC Beisamoun cremation practices appear at the end of a local thousand-year tradition of burying the dead within the settlement (at least a selected part of the dead). This is to say a redefinition of the place of the dead in the village and in society. Indeed, in the subsequent Pottery Neolithic period (especially in the earliest Yarmukian phase), in general, burials are very scarce in the Southern Levant [121, 122], attesting to a clear change in the treatment of the dead and a possible shift in beliefs. According to the data available from Beisamoun, this shift in adult funerary handling could have taken place before the end of the FPPNB/PPNC, approximately at 6500 cal BC.

## Supporting information

**S1 Fig. Sketches of the successive *décapages* at Beisamoun, Locus 338 that resulted in the exposure of human remains.** https://doi.org/10.7794/p6w6-f483.
(PDF)

**S1 Table. Comprehensive inventory and description of the human remains found in Locus 338, Beisamoun.** https://doi.org/10.7794/13jm-9k62.
(PDF)

## Acknowledgments

Simon Dorso and Caroline Masset were patient enough to excavate the cremation pit with all the meticulous attention required. Cécile Berton and Rachel Jadaud participated in the preparation of the bones (cleaning, restoration, refitting). The Ecole Biblique et Archéologique Française de Jérusalem generously hosted us during part of the lab work; we are especially grateful to J.-B. Humbert who provided assistance since the beginning of the project and a warm welcome. We are also indebted to the Centre de Recherche Français à Jérusalem which curates the material and have always received us warmly and provided the best working conditions. Alexandra Legrand, Yolanda Fernandez-Jalvo and Ferran Borrell offered us advice concerning the striae observed on the calcaneus and the projectile point embedded in the scapula. We offer our sincere thanks to: Luis Teira (Universitad de Cantabria), Alexandra Legrand (USR 3225), Rémi Brageu (LAMS) for their excellent micro and macroscopic photography; Denis Belœil and Fabrice Edon at the Service d'Imagerie médicale du Centre du Moulinet, Paris 13ème, who performed the X-ray and 3D imaging under the supervision of Laurent Buisson and Dr. Eric Auberton who kindly gave us their opinions on the injured shoulder; Isabelle Le Goff who provided us with important information and documents on cremation pyre-pit structures and combustion process; Nathalie Le Tellier-Becquart and Esther Magnière (USR3225) for generating the supplementary data' DOIs. Finally, we would like to warmly thank the reviewers, Scott D. Haddow, Ian Kuijt and a third anonymous reviewer, for their meticulous reviews and challenging questions which led to substantial improvements to the text.

## Author Contributions

**Conceptualization:** Fanny Bocquentin, Arlene Rosen, Hamoudi Khalaily, Liora Kolska Horwitz.

**Data curation:** Fanny Bocquentin, Marie Anton, Francesco Berna, Arlene Rosen, Hamoudi Khalaily, Omri Lernau, Liora Kolska Horwitz.

**Formal analysis:** Fanny Bocquentin, Marie Anton, Francesco Berna, Arlene Rosen, Harris Greenberg, Thomas C. Hart, Omri Lernau, Liora Kolska Horwitz.

**Funding acquisition:** Fanny Bocquentin, Arlene Rosen.

**Investigation:** Fanny Bocquentin, Marie Anton, Francesco Berna, Arlene Rosen, Harris Greenberg, Liora Kolska Horwitz.

**Methodology:** Fanny Bocquentin, Marie Anton, Francesco Berna, Arlene Rosen, Harris Greenberg, Liora Kolska Horwitz.

**Project administration:** Fanny Bocquentin, Hamoudi Khalaily.

**Supervision:** Fanny Bocquentin, Hamoudi Khalaily.

**Validation:** Fanny Bocquentin, Francesco Berna, Arlene Rosen, Hamoudi Khalaily, Liora Kolska Horwitz.

**Writing – original draft:** Fanny Bocquentin, Marie Anton, Francesco Berna, Arlene Rosen, Thomas C. Hart, Liora Kolska Horwitz.

**Writing – review & editing:** Fanny Bocquentin, Liora Kolska Horwitz.

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
