## [Decision Letter · Decision Letter 0]

2 Oct 2019

PONE-D-19-25674

Emergence of corpse cremation

during the Southern Levantine Pre Pottery Neolithic

PLOS ONE

Dear Dr Bocquentin,

Thank you for submitting your manuscript to PLOS ONE. After careful consideration, we feel that it has merit but does not fully meet PLOS ONE’s publication criteria as it currently stands. Therefore, we invite you to submit a revised version of the manuscript that addresses the points raised during the review process.

all comments must be addressed.

We would appreciate receiving your revised manuscript by Nov 16 2019 11:59PM. To enhance the reproducibility of your results, we recommend that if applicable you deposit your laboratory protocols in protocols.io, where a protocol can be assigned its own identifier (DOI) such that it can be cited independently in the future. For instructions see: http://journals.plos.org/plosone/s/submission-guidelines#loc-laboratory-protocols

We look forward to receiving your revised manuscript.

Kind regards,

Peter F. Biehl, PhD

Academic Editor

PLOS ONE

**Journal Requirements:**

2. In your manuscript, please provide additional information regarding the specimens used in your study. Ensure that you have reported specimen numbers and complete repository information, including museum name and geographic location.

For more information on PLOS ONE's requirements for paleontology and archaeology research, see "" ext-link-type="uri" xlink:type="simple">https://journals.plos.org/plosone/s/submission-guidelines#loc-paleontology-and-archaeology-research.""

3. We note that  Figure(s) 1 in your submission contain [map/satellite] images which may be copyrighted. All PLOS content is published under the Creative Commons Attribution License (CC BY 4.0), which means that the manuscript, images, and Supporting Information files will be freely available online, and any third party is permitted to access, download, copy, distribute, and use these materials in any way, even commercially, with proper attribution. For these reasons, we cannot publish previously copyrighted maps or satellite images created using proprietary data, such as Google software (Google Maps, Street View, and Earth). For more information, see our copyright guidelines: http://journals.plos.org/plosone/s/licenses-and-copyright.

a) You may seek permission from the original copyright holder of Figure(s) [#] to publish the content specifically under the CC BY 4.0 license.  

**Additional Editor Comments (if provided):**

Your manuscript has now been seen by two referees, whose comments are appended below. You will see from these comments that while the referees find your work of potential interest, at least one reviewer has raised substantial concerns that must be addressed. In light of these comments, we cannot accept the manuscript for publication, but would be interested in considering a revised version that addresses these serious concerns.

We hope you will find the referees' comments useful as you decide how to proceed. Should presentation of further data and analysis allow you to address these criticisms, we would be happy to look at a substantially revised manuscript. However, please bear in mind that we will be reluctant to approach the referees again in the absence of major revisions.

**Comments to the Author**

1. Is the manuscript technically sound, and do the data support the conclusions?

Reviewer #1: Yes

Reviewer #2: Yes

2. Has the statistical analysis been performed appropriately and rigorously? 

Reviewer #1: N/A

Reviewer #2: Yes

3. Have the authors made all data underlying the findings in their manuscript fully available?

Reviewer #1: Yes

Reviewer #2: No

4. Is the manuscript presented in an intelligible fashion and written in standard English?

Reviewer #1: Yes

Reviewer #2: No

5. Review Comments to the Author

Reviewer #1: This is an excellent paper that pulls together a number of interdisciplinary approaches to the study of what appears to be the earliest example of intentional cremation in the Near East. I recommend that it be published in PLoSONE with some minor revisions. My only major concern is that the authors have not made an entirely convincing case that the individual was cremated as a fresh corpse, and that alternative interpretations have not been sufficiently/explicitly examined. Please see additional comments and text edits in the attached pdf.

Reviewer #2: General comments

From a technical standpoint this is a robust and important study. I am strongly supportive of the publication of these materials. Having said this, there are significant revisions that need to take place before this paper is acceptable for publication.

Presentation and description of recovered materials

First, the authors need to reengage with and make the case for their interpretation of Locus 338 being the primary location of cremation (=cremation in the pit). They need to make a much stronger argument that considers the small physical scale of the feature and the practically of human cremation. At the moment the authors present a somewhat disjointed position, with surprisingly little discussion focused on taphonomy of the pit that is framed by other archaeological or ethnographic literature on cremation (see cited references below) or an understanding of the mechanics of fire. They argue that this was a small pit, 80 cm wide and 60 cm deep, and with a thin lining, perhaps one cm in thickness, with the bones of a single individual found in bottom 40 cm basis. The critical questions are 1) did the cremation take place inside of the basin, as they argue, or near the pit and the bone material later pushed in?, and 2) how much time passed between biological death of the individual and the act of cremation?

Their argument for primary context appears to be largely based on “some anatomical coherence” (Pg. 6.) and they argue that the burial found in locus 338 was burned in place. They argue that the cremated individual was probably seated, possibly fell forward, presumably over an open pit. Moreover, they argue that a few articulated bones, such as the phalanges of the right thumb, support argues for a “short delay between death and cremation”. In the final discussion they argue (Second sentence, Discussion, Pg. 16) “What remains of the deceased allows us to affirm that it was a relatively fresh corpse that has been burnt”. Confusingly they also argue in the conclusion (second paragraph, Pg. 17) that “This quantity of individuals packed in structures not larger (about 1 m diameter) than locus 338 of Beisamoun convinced the authors that that the deceased must have been cremated already skeletonized or at least decomposed (some joints are indeed still articulated).”

This needs to be pushed further. Depending upon conditions of humidity and temperature, defleshing as they argue can take months to years. This is not a “relatively fresh corpse”. I am not entirely convinced by the authors argument, and urge the authors to make a much stronger argument that draws upon more up to date literature as well ethnographic accounts of cremation. The pattern described here may well reflect cremation of a partially defleshed skeleton, but this takes years in dry and sandy conditions (think Near East, 8,200 kya). Moreover, can a skeletonized individual, partially defleshed, be seated? In addition to the literature the authors cite there is a significant body of literature (see references below) that highlights several physical aspects to creation: cremation requires more fuel that researchers assume, cremation requires good air flow and oxygen (and is therefore usually physically placed above the ground) and creation is, well, actually more complicated that researchers assume. From my reading the simplest explanation is that Locus 338 provides the remains of a cremated individual that was partially defleshed over multiple years, with the remains perhaps in a cloth bundle, burned above the pit, which then fell into the pit, and then at some point later (less than a week, possibly the same day) the additional fauna and other materials were swept into the shallow basin. The authors need a simple, cohesive narrative, such as the above, that brings this all into perspective. At the moment the current presentation is somewhat disjointed, inconsistent, and needs to be reframed in a more focused way.

Second, the core presentation of the skeletal analysis is inventive, visual (for example Fig. 6), and robust. At the same time the authors need to reengage with some of their core assumptions, making clear that these are interpretations and placing this argument within existing literature. For example, the authors spend a lot of time documenting the colour of burned bone materials (e.g., Fig, 4, which is framed around a colour gradation from black to white) but don’t directly explicitly make the case that different colour bone reflects specific different combustion history, accounting for variation in conditions such as fuel, burn time, and the amount of oxygen. Some of this discussion treats bone colouration in a very simplistic fashion, and with limited consideration of variation. Moreover, do all researchers identify visual color differences the same way?. What are the difference between Grey, Grey-White? Does this really reflect different burning environments? As a reader I am not convinced that this means anything. Is the entire bone in question Grey, or are parts of the bone Grey, with others being Grey-White, and others Grey-Black? (assuming we can agree what each of these is….). Again, these are good researchers, but some of this is being presented in an overly neat package that does not recognize variation. There are methodological and interpretative assumptions here that need to be made clear.

Third, I think the authors need to develop at least one robust paragraph that focused on excavation methods, data recovery, and documentation. Based on the skeletal analysis I am convinced that this was an excellent and professional excavation. The audience, however, needs to understand how Locus 338 was excavated, the methods used, the documentation process. None of this is directly presented, at least by section 3.2. A single substantial paragraph should address this.

Fourth, this paper is in need of editing as the presentation is at times rather cumbersome and confusing (Eg. Top two sentences of page 2. Pg. 5. “If an apparent anatomical anarchy is visible at first,..” ) and the authors need to clearly define their terms. In different sections in the paper the authors interchangeably use the term pit, cremation pit, structure, pyre structure, feature to talk about Locus 338. The authors need to select one descriptive term and the stick with it.

Fifth, at times the authors make some bold statements, that are only supported in a minor way, or in other cases, presented in a way that is very confusing.

Sec. 342. Pg 8 The authors state “The south, east and west walls show significant rubefaction indicating that the cremation was carried out on site.” What does this mean? The authors need to consider that this journal is focused on a general educated audience, so please slow down, take your time, and be clear.

Pg. 11. Last paragraph. “We observed five striae on the posterolateral face of the posterior talar facet of the right calcaneus (though the posterior talar facet was too fragmentary to be observed fully).” The general readers are not going to follow this.

“The practice of wrapping corpses has been practiced in this area since the Natufian period and is quite common in Neolithic contexts all over the Levant [e.g. 13, 46]. I agree that this is likely the case, but the authors are clearly overstating this. We have bits of evidence to support this argument, but these are small data points, not a conclusive understanding. Please rescale this argument.

Again, I remain supportive of the publication of this interesting work.

Cited literature

Thompson, T (ed.). 2015. The Archaeology of Cremation: Burned Human Remains in Funerary Studies [Paperback]. Oxbow books.

Schmidt, C. W. and S. A. Symes, 2008. The Analysis of Burned Human Remains, Elsevier.

Cerezo-Roman, A. Wessman, and H. Williams (eds.) 2017. Cremation and the Archaeology of Death. Oxford University Press.

6. PLOS authors have the option to publish the peer review history of their article (what does this mean?). If published, this will include your full peer review and any attached files.

Reviewer #1: Yes: Scott D. Haddow

Reviewer #2: No

---

## [Author Response · Author response to Decision Letter 0]

10 Jan 2020

We have answered all comments into two different files already uploaded.

---

## [Decision Letter · Decision Letter 1]

8 May 2020

PONE-D-19-25674R1

Emergence of corpse cremation during Pre Pottery Neolithic in the Southern Levant

PLOS ONE

Dear Dr Bocquentin,

Thank you for submitting your manuscript to PLOS ONE. After careful consideration, we feel that it has merit but does not fully meet PLOS ONE’s publication criteria as it currently stands. Therefore, we invite you to submit a revised version of the manuscript that addresses the points raised during the review process.

Please address these minor revisions/corrections.

We would appreciate receiving your revised manuscript by Jun 22 2020 11:59PM. To enhance the reproducibility of your results, we recommend that if applicable you deposit your laboratory protocols in protocols.io, where a protocol can be assigned its own identifier (DOI) such that it can be cited independently in the future. For instructions see: http://journals.plos.org/plosone/s/submission-guidelines#loc-laboratory-protocols

We look forward to receiving your revised manuscript.

Kind regards,

Peter F. Biehl, PhD

Academic Editor

PLOS ONE

Academic editor additional comments;

Your manuscript has now been seen again by the two referees, whose comments are appended below. You will see from these comments that while all major comments have been addressed the referees suggest minor revisions which will improve the manuscripts. As these are minor revisions we hope you can address them quickly and resubmit.

Reviewers' comments:

Reviewer's Responses to Questions

**Comments to the Author**

1. If the authors have adequately addressed your comments raised in a previous round of review and you feel that this manuscript is now acceptable for publication, you may indicate that here to bypass the “Comments to the Author” section, enter your conflict of interest statement in the “Confidential to Editor” section, and submit your "Accept" recommendation.

Reviewer #1: (No Response)

Reviewer #2: (No Response)

2. Is the manuscript technically sound, and do the data support the conclusions?

Reviewer #1: Partly

Reviewer #2: Yes

3. Has the statistical analysis been performed appropriately and rigorously? 

Reviewer #1: Yes

Reviewer #2: Yes

4. Have the authors made all data underlying the findings in their manuscript fully available?

Reviewer #1: No

Reviewer #2: Yes

5. Is the manuscript presented in an intelligible fashion and written in standard English?

Reviewer #1: Yes

Reviewer #2: Yes

6. Review Comments to the Author

Reviewer #1: The paper is considerably improved in its current version and I still believe this is an important and generally well-documented piece of research. However, I am disappointed that the authors persist in playing down or ignoring entirely the increasing evidence from other sites in the Neolithic Near East for delayed burial (involving either complete desiccation of the corpse or some form of soft tissue reduction, i.e. partial decomposition), especially in the discussion section where the last set of comments were completely ignored.

Regarding desiccation, under the right environmental conditions, or with intentional effort, human remains can desiccate in parts of the world that would not be considered "deserts". There are ethnographic examples from Papua New Guinea and other tropical environments (where corpses are smoked), for example, and even within Mediterranean coastal regions such as Sicily, where the well-known Capuchin mummies at Palermo desiccated naturally.

As with reviewer #2, I am skeptical about the feasibility of successfully cremating a fresh corpse in such a small and poorly ventilated space - both in terms of fuel load required and proper ventilation (regardless of whether or not the pit opening was left uncovered). I would find their argument more convincing if they acknowledged the possibility that the corpse may have been already partially decomposed, as it would likely require less fuel; proper ventilation may have been less of an issue as well, and would still accommodate the observation that joint articulations seem to have been maintained. They use the term "relatively fresh corpse", but perhaps they need to define what they mean by "relatively". It may be that we, as reviewers, and the authors are not so far apart after all on the possibility of a period of delay between death and cremation?

Regarding data availability (question 4 above), as mentioned in my comments on the first draft, it would have been nice to see more images of the burnt bones themselves so that one could independently assess the level of heat alterations to bone in the context of differences between dry and fleshed bone. I do understand however that providing images of every single bone is not feasible.

Reviewer #2: Author: Fanny Bocquentin

Title: Emergence of Corpse Cremation during the Southern Levantine Pre Pottery Neolithic

Journal: PLOS-ONE

File number: PONE-D-1925674

Rec: Accepted with minor revisions.

This is now an excellent paper, and should be published with minor revisions. The authors have made a good effort to address my concerns with the original version. I am supportive of the publication of this paper subject to addressing the issues raised below.

Reviewer 2: Ian Kuijt

General comments

First, while this is a great methodological paper, the paper suffers from having a very poor introduction. The introduction meanders and does not really tell the readers what the main focus of the paper will be….and then the paper kind of gets lost in history and symbolism. The authors need to identify the main focus of the paper. Is it strictly focused on demonstrating that this earliest occurrence of intentional primary cremation from the Near East? If so, this is a wasted opportunity. This is excellent research, well-illustrated and presented, but the authors do not step back from the trees and talk about the forest. I urge the authors to think about the big issues they identify at the end of the paper, in some ways using these themes to introduce the paper. The introduction and abstract are the most important ways you can draw in the readers, so these need to be crisp, focused on major evolutionary issues, and digestible to the readers.

Second, the paper does a poor job of defining the dates of the periods in question. For example, the PPNC / final PPNB is mentioned on page one, but not with specific details. It is mentioned in the abstract, but this also needs to be in the body of the text.

Third, I would urge the authors to go back and add a sub-heading to introduce the site of Beisamoun and reorganize the section Chrono-Cultural Context. I think the original was more effective. Right now the site is introduced in a hidden way, buried in the middle of the section titled Chrono-Cultural Context. This is a poorly developed section, with the title hinting at a discussion of regional culture-history, and then in the middle of this section we are introduced to Beisamoun.

Fourth, the authors have added a much needed section on excavation methods (sections 2.1., 2.2.). These are excellent and really, really, make this an outstanding paper.

Fifth, section 2.6.3. deals with the burning event. This is a good section. My only suggestion is to add more clarity, if you can, about the placement of the wood and body. You talk about fuel and other great topics, but I am struggling to understand what you see as the original position of the fuel and body. For example, do you believe that the 60 cm deep pit was filled with wood, up to and above the ground surface, and then the fire was started? Given the need for oxygen this makes sense, for it is all about oxygen. In another section you state that the person may have been seated in the 60 cm high pit. If so, was fuel placed around the body? If this is your argument then place find a place, probably as the lead sentence of one of your paragraphs to state this up front. Help you reader understand your overall view, and then dive into the details.

Specific comments:

Fig. 1. The names of multiple sites are misspelled on the map and need to be corrected.

Tell el-Kerkh is the correct published spelling, not Tell-el-Kherk as listed on Fig. 1.

Yarim Tepe II is the correct published spelling, not Yarim tepe II as listed on Fig. 1

‘Ain Ghazal is the correct published spelling, not Aïn Ghazal as listed on Fig. 1. Sorry, but M. Sauvage, CNRS, needs to correct this figure before publication.

Pg. 2., first sent, first paragraph. “The treatment of the dead during the Neolithization process in the Near East was a complex process, part of the cognitive and symbolic world that guided the economic and dietary shift from hunting and gathering to agro-pastoralism [e.g. articles in 1].” This suggests that the treatment of the dead “guided” the shift from hunting and gathering to agro-pastoralism. I would urge the authors to reconsider this argument, or perhaps it is a misstatement.

Pg. 3., first paragraph “…as the condemnation of the funeral building itself by fire.” This is poor wording. Might I suggest “the burning of the funerary building”. Condemnation is not a clear way to present this.

Pg. 3., first paragraph, and throughout the entire paper, “…site of Tell Ain el Kherk in a …”. As correctly cited in the references, the name of this site is Tell el-Kerkh (see Tsuneki 2012). To meet the publication standards of this journal it is critical that the authors employ the correct, published spelling of all sites in the text, references and figures.

Pg. 4., end of second paragraph “They are currently kept at the French National

Research Center in Jerusalem (CRFJ) and they will be handed over to the IAA after publication. In the meantime, they are available for review upon request to the CRFJ and the corresponding author of the current article.” This should be in a foot or endnote. This does not advance discussion: it is an administrative aside.

Pg. 4., first sentence of section 2.1. “As for all features…”. Should be corrected to “As with all features….”

The paper reads well, but as with all essays at this point of production, there are typos and grammatical issues that need to be cleared up before publication.

Cited literature

Tsuneki, A. 2012. Tell el-Kerkh as a Neolithic Mega Site. Orient. Vol. XLVII: 29-66

7. PLOS authors have the option to publish the peer review history of their article (what does this mean?). If published, this will include your full peer review and any attached files.

Reviewer #1: Yes: Scott D Haddow.

Reviewer #2: Yes: Ian Kuijt

---

## [Author Response · Author response to Decision Letter 1]

9 Jun 2020

All responses were made in the attached document "response to reviewers"

---

## [Editor Report · Decision Letter 2]

16 Jun 2020

Emergence of corpse cremation during the Pre-Pottery Neolithic of the Southern Levant: A multidisciplinary study of a pyre-pit burial

PONE-D-19-25674R2

Dear Dr. Bocquentin,

We’re pleased to inform you that your manuscript has been judged scientifically suitable for publication and will be formally accepted for publication once it meets all outstanding technical requirements.

Kind regards,

Peter F. Biehl, PhD

Academic Editor

PLOS ONE
---

## [Editor Report · Acceptance letter]

9 Jul 2020

PONE-D-19-25674R2 

Emergence of corpse cremation during the Pre-Pottery Neolithic of the Southern Levant: A multidisciplinary study of a pyre-pit burial 

Dear Dr. Bocquentin:

I'm pleased to inform you that your manuscript has been deemed suitable for publication in PLOS ONE. Congratulations! Your manuscript is now with our production department. 

Kind regards, 

on behalf of

Dr. Peter F. Biehl 

Academic Editor

PLOS ONE